# Perfluoroalkyl substance pollutants activate the innate immune system through the AIM2 inflammasome

Li-Qiu Wang[1], Tao Liu[1], Shuai Yang[1], Lin Sun[2], Zhi-Yao Zhao[1], Li-Yue Li[2], Yuan-Chu She[1], Yan-Yan Zheng[1], Xiao-Yan Ye[2], Qing Bao[2], Guang-Hui Dong [3], Chun-Wei Li[2 ✉] & Jun Cui [1 ✉]

Perfluoroalkyl substances (PFAS) are widely used in various manufacturing processes. Accumulation of these chemicals has adverse effects on human health, including inflammation in multiple organs, yet how PFAS are sensed by host cells, and how tissue inflammation eventually incurs, is still unclear. Here, we show that the double-stranded DNA receptor AIM2 is able to recognize perfluorooctane sulfonate (PFOS), a common form of PFAS, to trigger IL-1β secretion and pyroptosis. Mechanistically, PFOS activates the AIM2 inflammasome in a process involving mitochondrial DNA release through the $Ca^{2+}$-PKC-NF-κB/JNK-BAX/BAK axis. Accordingly, $Aim2^{-/-}$ mice have reduced PFOS-induced inflammation, as well as tissue damage in the lungs, livers, and kidneys in both their basic condition and in an asthmatic exacerbation model. Our results thus suggest a function of AIM2 in PFOS-mediated tissue inflammation, and identify AIM2 as a major pattern recognition receptor in response to the environmental organic pollutants.

[1] MOE Key Laboratory of Gene Function and Regulation, State Key Laboratory of Biocontrol, School of Life Sciences, Sun Yat-sen University, Guangzhou, Guangdong, China. [2] Department of Otolaryngology, The First Affiliated Hospital, Sun Yat-sen University, Guangzhou, Guangdong, China. [3] Guangdong Provincial Engineering Technology Research Center of Environmental Pollution and Health Risk Assessment, Department of Occupational and Environmental Health, School of Public Health, Sun Yat-sen University, Guangzhou, Guangdong, China. ✉email: hi_chunwei@aliyun.com; cuij5@mail.sysu.edu.cn

Perfluoroalkyl substances (PFAS) are widely used in industrial manufacturing processes and consumer products due to their joint hydrophobic and oleophobic properties[1,2]. These compounds are detected in many routinely used items during our daily life, such as coats, papers, food contact materials, cleansers, paints, etc. PFAS are hard to be degraded under natural environmental conditions and can also be detected in the environment, plants, and wildlife[2,3]. Human populations are exposed to PFAS via ingestion of drinking water and food, inhalation of air and dust, and contact with contaminated media[2,4]. Emerging evidence has suggested an accumulation of PFAS in the human body is associated with adverse health effects, including immune-related health conditions (like allergic diseases, infection, and vaccine response), metabolic dysregulation (like nonalcoholic fatty liver disease, chronic kidney diseases), and neurodevelopmental delays[5–9]. Animal and in vitro cell experiments also demonstrate that PFAS have immunotoxic and oxidative effects on cells and tissues, leading to severe inflammation in multiple organs (like liver, kidney, lung, and nervous system)[10–15]. Among the various PFAS, perfluorooctane sulfonate (PFOS; $C_8F_{17}SO_3^-$) is the most common type studied in relation to human health and was added to the Stockholm Convention's list of globally restricted Persistent Organic Pollutants in 2009 under United Nation Environment Programme.

Several studies have indicated that oxidative and inflammatory responses are involved in PFOS-induced cell injury and tissue damage. PFOS induces oxidative stress via inhibiting antioxidant factors (such as superoxide dismutase, lysozyme) or promoting reactive oxygen species in hepatocytes or endothelial cells, leading to cell death[11,13,14,16,17]. In addition, PFOS enhances the expression of proinflammatory cytokines (IL-1β, TNF-α, and IL-6) in macrophages[18,19]. However, most experimental results about PFOS are based on descriptive studies and the mechanisms underlying PFOS-induced tissue inflammation remain poorly understood. One critical question is how PFOS is recognized in the cells and triggers cellular inflammatory responses.

Pattern recognition receptors (PRRs) are a broad family of proteins expressed by various cells of the innate immune system, recognizing conserved molecular moieties commonly associated with pathogens and environmental factors. PRRs recognize specific molecular patterns (e.g., protein, RNA, and DNA) of microorganisms, referred to as pathogen-associated molecular patterns (PAMPs), activating a series of innate immune pathways. Unlike PAMPs from microbes, the environmental toxins or pollutants are usually not recognized directly by PRRs, but they may induce danger signals released from injury cells, typically known as danger-associated molecular patterns (DAMPs); consequently, PRRs can sense DAMPs and trigger proinflammatory responses. Nucleotide-binding and oligomerization domain (NOD)-like receptors (NLRs), including NOD1, NOD2, and NLRP3, belongs to major intracellular PRR families which regulate inflammatory responses. Previous reports have shown that silica crystals could induce lysosomal disruption, which then activates NLRP3 inflammasome, and leads to lung inflammation[20,21]. Other studies also showed that NLRP3 inflammasome can be activated by other environmental pollutants, such as particulate matters (PM2.5, PM10), carbon black nanoparticles, and nickel[22–26]. However, the host sensing mechanisms for most environmental factors are still largely unknown.

Given the facts that the inflammasome is a key sensor for environmental factors, and PFOS can promote IL-1β production in macrophages, we hypothesize that inflammasome may potentially recognize PFOS or PFOS-induced DAMPs and then result in inflammatory cytokines release, leading to tissue damage.

Here we show that PFOS can induce AIM2 inflammasome-dependent IL-1β production and pyroptosis in macrophages, and this process is mediated by PFOS-induced mitochondrial DNA (mtDNA) release. Moreover, Aim2-deficient (Aim2−/−) mice are protected from PFOS-induced tissue inflammation and exacerbation. Interestingly, we observe that PFOS-mediated inflammatory response is independent of NLRP3 inflammasome, based on the results from NLRP3 knockout cell lines and Nlrp3−/− mice. Collectively, our results suggest a link between environmental organic pollutants (e.g., PFOS), intracellular danger signals (mtDNA) and sensor (AIM2 inflammasome) to the host effect (cellular inflammation and tissue damage), which is the essential mechanism triggering the PFOS-associated inflammatory diseases.

## Results

**PFOS induces inflammasome activation.** To investigate the potential effects of PFOS in tissue inflammation in vivo, wild type (WT) C57BL/6J mice were i.p. injected with different doses of PFOS (i.e., acute exposure, 5, 15, or 25 mg/kg body weight per day for 5 days; chronic exposure, 0.066 mg/kg body weight per day for 30 days), which was modified by the reported methods[27–29]. Compared to control group, increased inflammatory cell infiltration and tissue damage were observed in the liver, lung, and kidney of PFOS-treated mice (Supplementary Figs. 1a and 2a). The proinflammatory cytokines IL-1β, TNF-α, and IL-6 were significantly increased in the serum, peritoneal fluids, liver, lung, and kidney in the PFOS-treated mice, compared to those in control mice (Supplementary Figs. 1b–d and 2b–d). These results demonstrate that both the acute and chronic PFOS exposure induce inflammatory cytokine production and tissue injury in vivo, which are in line with the previous findings[11,12,15].

To gain insight into the mechanisms of inflammation induced by PFOS stimulation, we treated bone marrow-derived macrophages (BMDMs) with PFOS. PFOS could significantly induce the expression and secretion of IL-1β and cell death in a concentration- or time-dependent manner (Fig. 1a and Supplementary Fig. 3a). It also induced the mRNA levels and the release of TNF-α and IL-6 (Supplementary Fig. 4a, b), indicating that PFOS could induce the activation of NF-κB signaling pathway, which is consistent with the previous report[19]. Next, we investigated the mechanism of underlying PFOS-induced NF-κB signaling activation. In BMDMs, PFOS exposure triggered the phosphorylation of NF-κB p65 subunit and the degradation of NF-κB inhibitor alpha (IκBα) (Supplementary Fig. 4c). The increased $Ca^{2+}$ in cytosolic compartment ($[Ca^{2+}]c$) is essential for downstream calcium-dependent signaling pathway activation (such as protein kinase C), to mediate NF-κB signaling activation. Since $Ca^{2+}$ signaling can be triggered by the release of $Ca^{2+}$ from intracellular endoplasmic reticulum (ER) storage[30–34], we analyzed the effect of PFOS on $[Ca^{2+}]c$ and found that $[Ca^{2+}]c$ and the protein level of ER stress monitor Bip were significantly increased in PFOS-treated BMDMs (Supplementary Fig. 4d). To further determine the involvement of $[Ca^{2+}]c$ and protein kinase C (PKC) in PFOS-triggered NF-κB signaling activation, we pretreated BMDMs with $Ca^{2+}$ chelator (BAPTA-AM) or PKC inhibitor chelerythrine chloride (Ch-chloride) before PFOS treatment, and found that phosphorylation of p65, the degradation of IκBα and proinflammatory cytokines (IL-1β, TNF-α, and IL-6) were evidently inhibited in the presence of $Ca^{2+}$ chelator or PKC inhibitor (Supplementary Fig. 4e–g). Collectively, our results showed that $Ca^{2+}$–PKC-dependent pathway is critical for PFOS-mediated NF-κB signaling activation, which led to the transcription and production of pro-IL-1β, TNF-α, and IL-6.

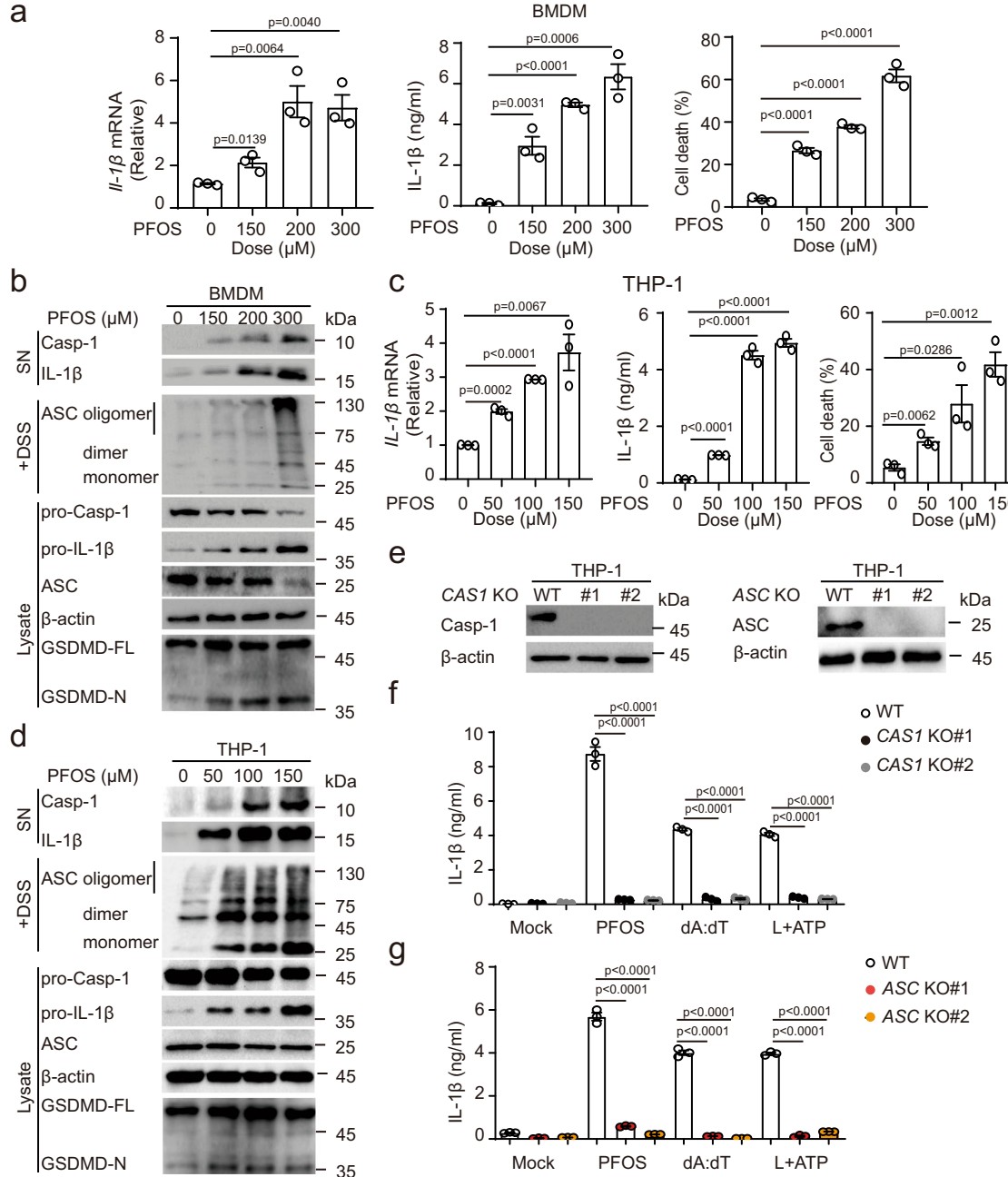

**Fig. 1 PFOS triggers caspase-1 activation and IL-1β secretion in macrophages. a, b** Bone marrow-derived macrophages (BMDMs) were treated with PFOS in indicated dose points for 6 h, cell lysates were collected to determine mRNA levels of *IL-1β*, and cell supernatants were collected to measure IL-1β secretion and cell death (**a**). Immunoblot analysis of of caspase-1 activation and IL-1β maturation in the supernatants, and ASC oligomerization, pro-caspase-1, pro-IL-1β, and GSDMD cleavage in the lysates of PFOS-treated BMDMs (**b**). **c, d** PMA-differentiated THP-1 cells (THP-1-derived macrophages) were stimulated with PFOS as indicated for 6 h. Cell lysates were harvested to analyze mRNA levels of *IL-1β*, and cell supernatants were collected to determine IL-1β and IL-18 production, and cell death (**c**). Immunoblot analysis of caspase-1 activation and IL-1β maturation in the supernatants, and ASC oligomerization, pro-caspase-1, pro-IL-1β, and GSDMD cleavage in the lysates of PFOS-treated THP-1-derived macrophages (**d**). **e** Knockout (KO) efficiency of *CASPASE-1* (*CAS1*) KO and *ASC* KO THP-1-derived macrophages were evaluated by immunoblot. **f, g** IL-1β secretion from wild type (WT), two clones of *CAS1* KO (KO#1: black plots, KO#2: gray plots) (**f**) or *ASC* KO (KO#1: red plots; KO#2: orange plots) (**g**) THP-1-derived macrophages was determined by ELISA. The cells were treated with PFOS (150 μM, 6 h) or poly (dA:dT) (2 μg/ml, 6 h) or pre-treated with LPS (200 ng/ml, 3 h) followed by ATP (5 mM, 6 h). In **a, c, f** and **g**, all error bars, mean values ± SEM, *P*-values were determined by unpaired two-tailed Student's *t* test of *n* = 3 independent biological experiments. For **b, d** and **e**, similar results are obtained for three independent biological experiments. Source data are provided as a Source data file.

Since the inflammasome activation requires two signals (priming signals for upregulating the transcriptional mRNA level of *IL-1β* and secondary signals to activate caspase-1 for IL-1β procession) for the maturation and secretion of IL-1β, we sought to determine the ability of PFOS to elicit inflammasome activation. Immunoblot analysis showed that PFOS could induce ASC oligomerization, caspase-1 activation, GSDMD cleavage, and IL-1β maturation in a concentration-dependent manner in BMDMs (Fig. 1b), indicating that PFOS triggers inflammasome activation to induce IL-1β secretion and pyroptosis in BMDMs.

Next, we sought to determine whether the similar effects existed in PFOS-treated human macrophages. We treated THP-1-derived macrophages with PFOS and confirmed that PFOS could also increase the mRNA level of *IL-1β*, protein secretion of IL-1β, and cell death of THP-1-derived macrophages in a concentration- or time-dependent manner (Fig. 1c and Supplementary Fig. 3b), as well as the expression and release of TNF-α and IL-6 (Supplementary Fig. 5a, b). Consistent with the observations in BMDMs, $Ca^{2+}$–PKC-dependent pathway was also responsible for PFOS-induced NF-κB signaling activation in THP-1-derived macrophages (Supplementary Fig. 5c–g). The serum level of PFOS in most human subjects ranges from 0.2 to 99.7 ng/ml[35,36], and this translates to about 200 nM concentration (PFOS's molecular mass is 538.22); then we treated THP-1-derived macrophages with 10–200 nM of PFOS, and found that lower concentrations of PFOS could also trigger the proinflammatory cytokines production and cell death in macrophages (Supplementary Fig. 6a, b). In line with that, PFOS-treated THP-1-derived macrophages also elicited inflammasome activation via triggering ASC oligomerization, cleavage of caspase-1, GSDMD, and maturation of IL-1β (Fig. 1d). This result was further confirmed in *CASPASE-1* knockout (*CAS1* KO) THP-1-derived macrophages (Fig. 1e), indicating that caspase-1 is involved in PFOS-induced secretion of IL-1β (Fig. 1f). Furthermore, PFOS-induced IL-1β secretion was completely abolished in *ASC* KO THP-1-derived macrophages (Fig. 1e, g). These findings suggest that PFOS-induced inflammasome activation depends on both caspase-1 and ASC. Hence, our results indicate that both human and mouse macrophages engage in inflammasome activation and pyroptosis in response to PFOS exposure.

**PFOS specifically activates AIM2 inflammasome**. We next sought to explore what kind of inflammasome was involved in the PFOS-induced production. NLRP3 and AIM2 are known to play critical roles in inflammasome activation upon a variety of cellular infection or stress. We observed that inhibition of NLRP3 with MCC950 only inhibited IL-1β secretion in response to the classical NLRP3 agonist ATP, but not the classical AIM2 agonist poly(dA:dT) or PFOS (Supplementary Fig. 7a). Consistent with this result, we found that the inflammasome activation induced by NLRP3 agonist ATP was completely abolished in *NLRP3* KO THP-1-derived macrophages (Fig. 2a, b). However, PFOS- or AIM2 agonist poly(dA:dT)-induced IL-1β secretion, caspase-1 activation, and IL-1β maturation were not affected in *NLRP3* KO THP-1-derived macrophages (Fig. 2a, b), suggesting that NLRP3 is not involved in PFOS-induced inflammasome activation. In contrast, in *AIM2* KO THP-1-derived macrophages, PFOS-induced IL-1β secretion, caspase-1 activation, and IL-1β maturation were completely abrogated (Fig. 2c, d). Similarly, we found that BMDMs from $Aim2^{-/-}$ mice but not $Nlrp3^{-/-}$ mice failed to induce caspase-1 activation, IL-1β maturation, and secretion after PFOS treatment (Fig. 2e, f). In addition, we determined the functions of other inflammasomes under PFOS stimulation by knocking down *NLRP1* or *NLRC4* in the THP-1-derived macrophages, and the results showed that PFOS-induced-IL-1β production was not dependent on NLRP1 or NLRC4 in THP-1-derived macrophages (Supplementary Fig. 7b, c). Furthermore, we observed that the PFOS-induced cell death was decreased in AIM2 inflammasome components deficient THP-1-derived macrophages and $Aim2^{-/-}$ BMDMs, indicating that AIM2 inflammasome activation contributes to PFOS-triggered cell death (Supplementary Fig. 8a–d). In line with that, PFOS-induced IL-1β production and cell death were apparently attenuated in *AIM2* KO THP-1-derived macrophages treated with low concentration (200 nM), suggesting that AIM2 plays an

important role in PFOS-induced chronic inflammasome activation (Supplementary Fig. 8e, f). Collectively, the above results indicate that AIM2 acts as a critical sensor for PFOS in inflammasome activation.

**PFOS induces cytosolic mitochondrial DNA accumulation**. AIM2 is a well-studied pattern recognition receptor that mainly senses DNA to induce inflammation[37,38]. Thus, we hypothesized that PFOS treatment might induce the release of host DNA into the cytoplasm as the secondary messenger to activate AIM2 inflammasome. We treated THP-1-derived macrophages with PFOS in a concentration-dependent manner and then extracted cytosolic DNA. We found that PFOS treatment induced the cytosolic mitochondrial DNA (mtDNA) accumulation (Fig. 3a). To further confirm this observation, we pulled down AIM2 protein and assessed relative enrichment of mtDNA versus genomic DNA (gDNA) after PFOS treatment. Our data showed that PFOS could not enhance the interaction between gDNA fragments and AIM2 (Fig. 3b), while mtDNA fragments had significantly increased interaction with AIM2 rather than NLRP3 after PFOS treatment (Fig. 3c and Supplementary Fig. 9a). NLRP3 activators (e.g., ATP, NIG) caused the release of oxidized mtDNA into the cytosol to interact with and activate NLRP3 inflammasome[39,40], so we assumed that non-oxidized mtDNA preferentially induces AIM2 inflammasome activation and oxidized-mtDNA preferentially activates NLRP3 inflammasome. We synthesized a 90-base-pair (bp) DNA fragment encompassing the D-loop region of mouse mtDNA with or without oxidation (8-OH-dGTP) to confirm this speculation. Normal mtDNA (non-oxidized mtDNA) and mtDNA containing 8-OH-dGTP (oxidized mtDNA) were transfected into BMDMs, and non-oxidized mtDNA-induced-IL-1β production was significantly decreased in $Aim2^{-/-}$ BMDMs, but not in $Nlrp3^{-/-}$ BMDMs (Supplementary Fig. 9b). Furthermore, oxidized mtDNA could still induce IL-1β release in $Aim2^{-/-}$ BMDMs rather than $Nlrp3^{-/-}$ BMDMs (Supplementary Fig. 9b). Hence, our results indicate that non-oxidized mtDNA induces inflammasome activation via preferential activation of AIM2, but not NLRP3. Taken together, these data suggest that PFOS induces the release of non-oxidized mtDNA into the cytosolic compartment to interact with AIM2 and activate AIM2 inflammasome.

To investigate the relationships between cytosolic mtDNA accumulation and inflammasome activation, we challenged THP-1-derived macrophages and BMDMs with ethidium bromide (EtBr) to generate $\rho^0$ cells with depletion of mtDNA[41]. We cultured THP-1-derived macrophages or BMDMs for 7 days in the presence of EtBr and generated the cells with ~80% loss of mtDNA (Fig. 3d). We found that the $\rho^0$-THP-1-derived macrophages or $\rho^0$-BMDMs displayed nearly completed elimination of the response to PFOS-induced inflammasome activation (Fig. 3e–h). These data indicate that PFOS treatment results in mitochondrial DNA accumulation in the cytosolic compartment and subsequent AIM2 inflammasome activation.

**Contribution of cyclophilin D in PFOS-induced mtDNA release**. Mitochondrion, an organelle of oxidative metabolism machinery, comprises a connected network containing multiple copies of mtDNA that highly compacted with the mitochondrial matrix. Recent studies have been shown that mitochondrial dysfunction can result in mtDNA release into the cytosol[42,43]. In addition, previous reports showed that PFOS induced toxic effects toward cardiomyocytes and hepatocytes, and caused hepatic damage through mitochondria-dependent pathway[16–19]. Therefore, we investigated whether PFOS caused mitochondrial dysfunction in macrophages. Firstly, we measured the mitochondrial

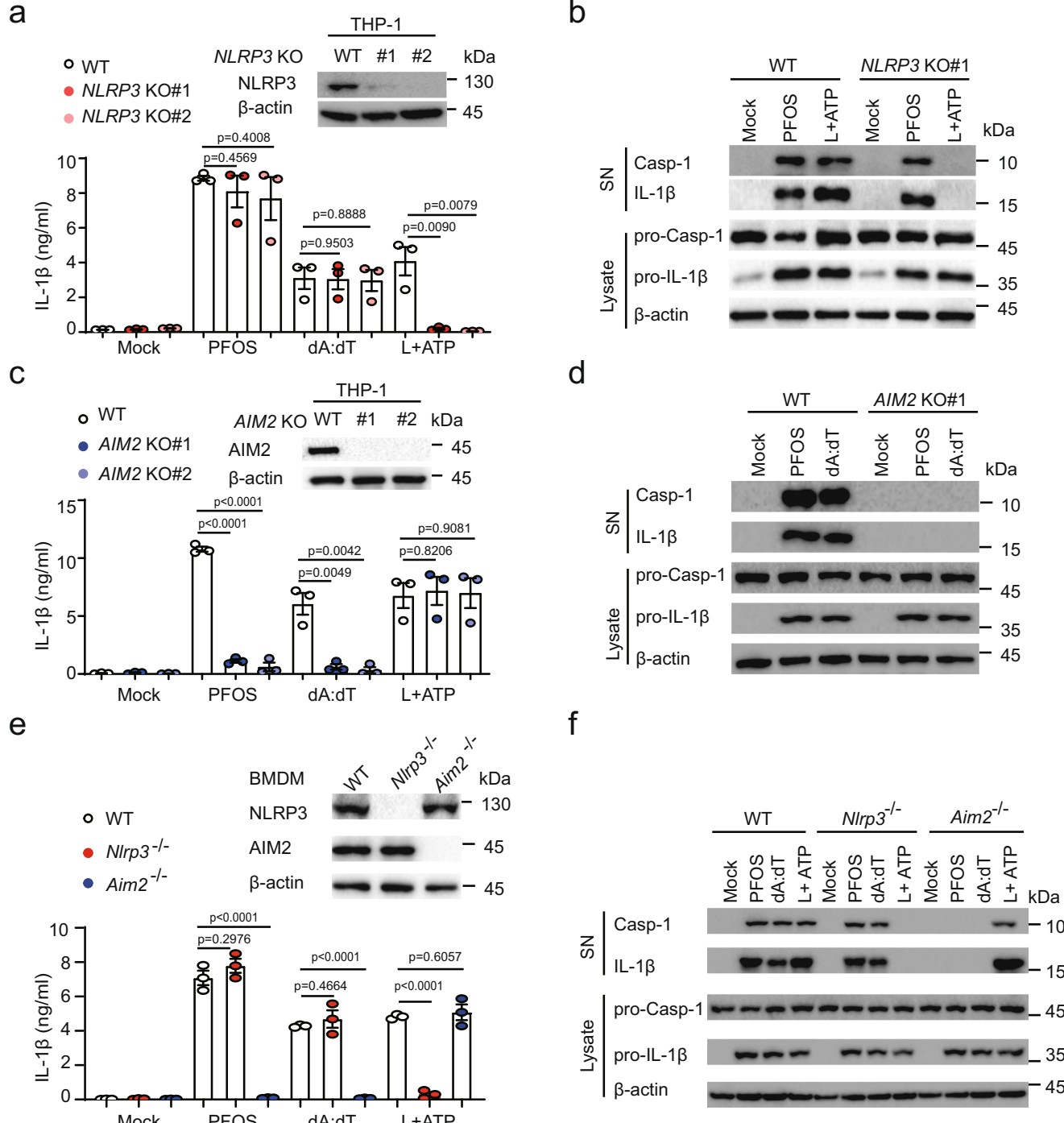

**Fig. 2 PFOS specifically activates AIM2 inflammasome. a–d** Wild type (WT) or indicated *NLRP3* knockout (KO#1: deep red plots; KO#2: light red plots) THP-1-derived macrophages (**a**, **b**) or *AIM2* KO (KO#1: deep blue plots; KO#2: light blue plots) THP-1-derived macrophages (**c**, **d**) were treated with PFOS (150 μM, 6 h), poly (dA:dT) (2 μg/ml, 6 h) or pre-treated with LPS (200 ng/ml, 3 h) before ATP (5 mM, 6 h). IL-1β release was measured in the supernatants by ELISA (**a**, **c**). Cell lysates and supernatants were harvested for immunoblot (**b**, **d**). **e**, **f** WT, *Nlrp3*$^{-/-}$ (red plots) or *Aim2*$^{-/-}$ (blue plots) Bone marrow-derived macrophages (BMDMs) were treated with indicated stimulations, and supernatants were collected to determine IL-1β production by ELISA (**e**), cell extracts and supernatants were collected for immunoblot (**f**). In **a**, **c** and **e**, all error bars, mean values ± SEM, *P*-values were determined by unpaired two-tailed Student's *t* test of *n* = 3 independent biological experiments. For **b**, **d** and **f**, similar results are obtained for three independent biological experiments. Source data are provided as a Source data file.

ROS production using MitoSOX fluorescence by flow cytometry. Surprisingly, while LPS plus ATP treatment heightened mtROS as expected[44], mtROS production showed upregulated trend with no statistically significant in PFOS-treated THP-1-derived macrophages and BMDMs (Supplementary Fig. 10a, b). In agreement

with this striking result, treatment of cells with Mito-TEMPO, a specific scavenger for mtROS, didn't affect IL-1β maturation and IL-1β secretion in both THP-1-derived macrophages and BMDMs in response to PFOS, but impaired ATP-induced NLRP3 inflammasome activation (Supplementary Fig. 10c–f). Thus, we

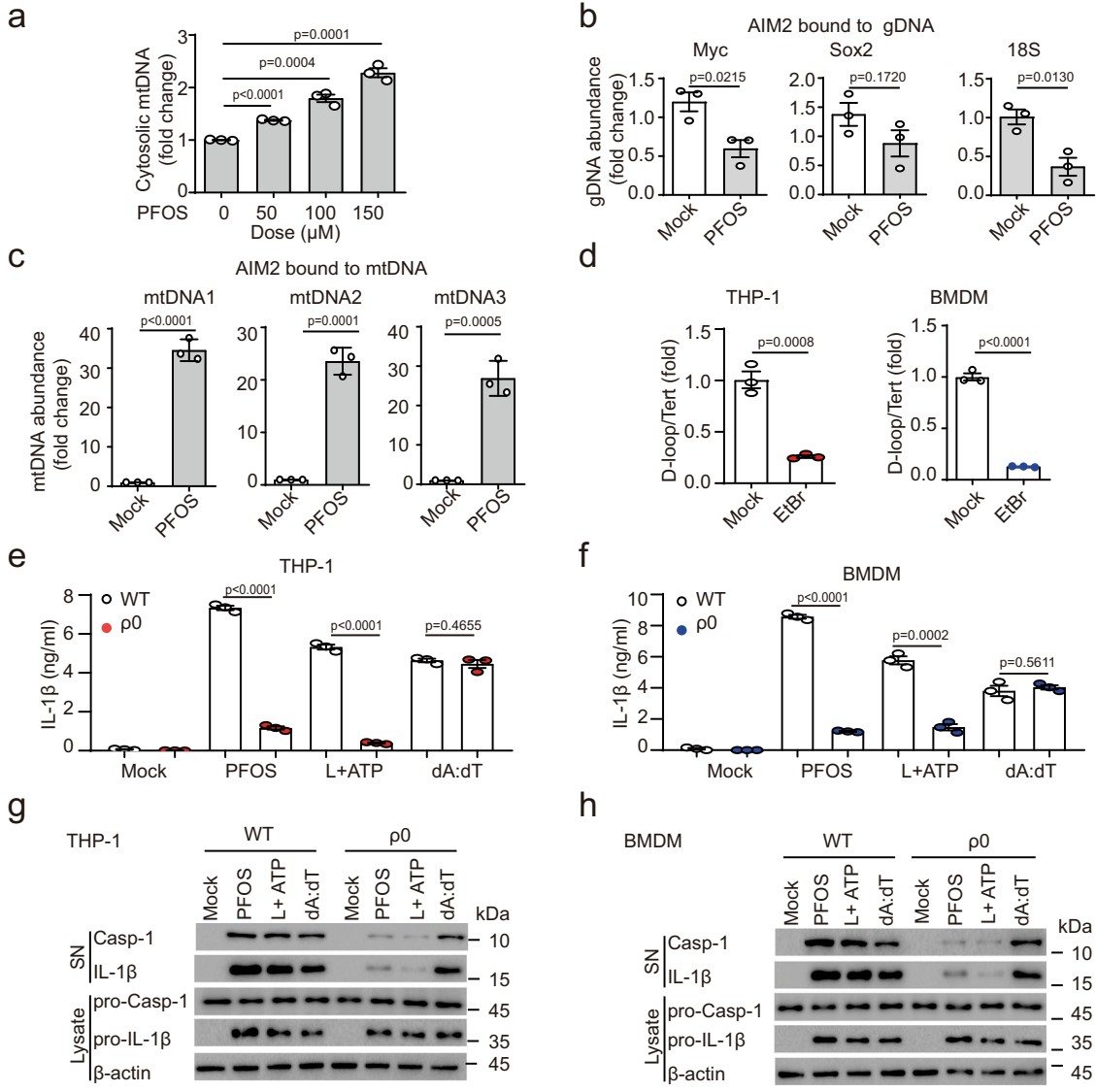

**Fig. 3 PFOS-induced mtDNA accumulation triggers AIM2 inflammasome activation. a** mtDNA amount from cytosolic extracts derived from THP-1-derived macrophages stimulated with PFOS was detected by qRT-PCR. **b, c** Relative enrichment of DNA in AIM2-pulldown material from Mock or PFOS (150 μM, 6 h) condition. qRT-PCR for three sets of primers that amplify fragments from different regions of the human genomic gDNA (**b**) or mtDNA (**c**). **d** mtDNA amount of total DNA extractions from THP-1-derived macrophages or bone marrow-derived macrophages (BMDMs) treated for 7 days without or with ethidium bromide (EtBr). **e–h** ELISA analysis of IL-1β secretion from supernatants of THP-1-derived macrophages (**e**, red plots) or BMDMs (**f**, blue plots) depleted of mtDNA and stimulated with PFOS (150 μM, 6 h), poly (dA:dT) (2 μg/ml, 6 h) or pre-treated with LPS (200 ng/ml, 3 h) followed by ATP (5 mM, 6 h). The cleavage of caspase-1 (p10) and the maturation of IL-1β (p17) in the cell supernatants or pro-Casp1 and pro-IL-1β in the cell lysates of THP-1-derived macrophages (**g**) or Bone marrow-derived macrophages (BMDMs) (**h**) depleted of mtDNA were determined by immunoblot. In **a–f**, all error bars, mean values ± SEM, P-values were determined by unpaired two-tailed Student's t test of n = 3 independent biological experiments. For **g** and **h**, similar results are obtained for three independent biological experiments. Source data are provided as a Source data file.

speculated that PFOS-induced AIM2 inflammasome activation was independent on mitochondrial ROS production. We further assessed the functional mitochondrial respiration in THP-1-derived macrophages and BMDMs using Mitotracker deep red staining by flow cytometry. Mitotracker deep red staining was significantly decreased in both THP-1-derived macrophages and BMDMs after PFOS treatment, indicating that PFOS impairs mitochondrial respiration (Supplementary Fig. 11a and Fig. 4a).

Mitochondrial dysfunction is generally linked with alterations in the ion potential of the inner membrane[39]. Therefore, we examined whether PFOS changed mitochondrial membrane potential with tetramethylrhodamine methyl ester (TMRM)

staining. PFOS stimulation led to diminishing TMRM staining in macrophages, indicating that PFOS could induce mitochondrial dysfunction (Supplementary Fig. 11b and Fig. 4b).

Cytochrome c (Cyt c), a protein normally resided in the mitochondrial intermembrane space, participates in the mitochondrial electron-transport chain[45]. It has been reported that Cyt c loss preceded mtDNA release[46]. Here we found that mitochondrial Cyt c was released into the cytosol in both PFOS-treated THP-1-derived macrophages and BMDMs (Supplementary Fig. 11c and Fig. 4c).

Cyt c could release from mitochondria into the cytosolic compartment during apoptosis [47,48], and we found that PFOS still

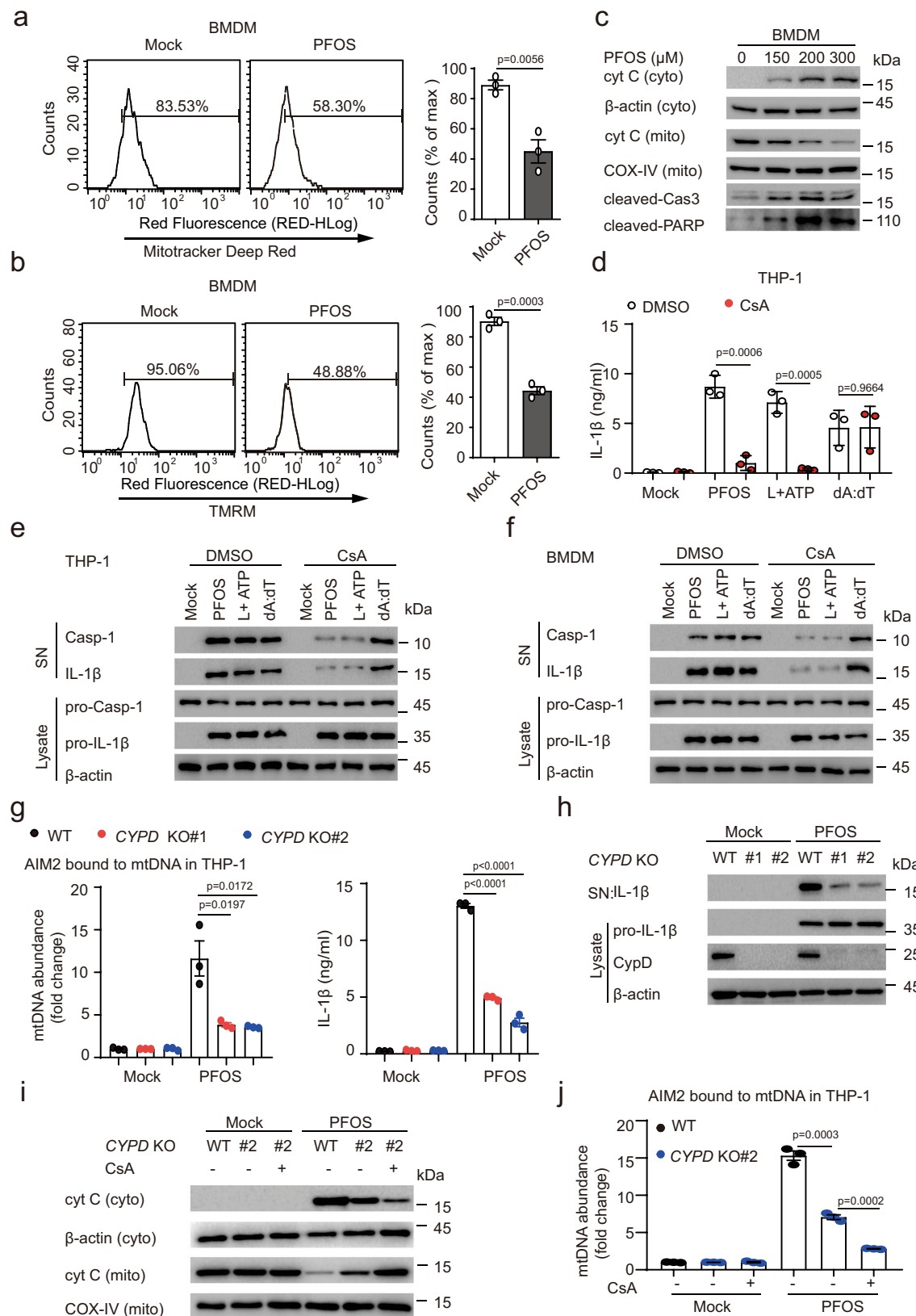

induced obvious cell death in AIM2 inflammasome components deficient cells (Supplementary Fig. 8), suggesting that PFOS triggers both pyroptosis and other types of cell death, including apoptosis. Hence, to determine whether PFOS could induce apoptosis, we detected the protein levels of cleaved caspase-3 and PARP in macrophages. PFOS treatment could apparently enhance caspase-3 and PARP cleavage in both THP-1-derived macrophages and BMDMs (Supplementary Fig. 11d and Fig. 4c). We further assayed the PFOS-induced apoptosis with Annexin V/ 7AAD staining by flow cytometry analysis, and observed that PFOS could trigger evident apoptosis in a dose-dependent manner (Supplementary Fig. 11e). These data indicate that PFOS

**Fig. 4 PFOS induces mitochondrial dysfunction and apoptosis in macrophages. a, b** Bone marrow-derived macrophages (BMDMs) were treated with PFOS (150 μM, 6 h), then the cells were stained with Mitotracker deep red (**a**) or tetramethylrhodamine methyl ester (TMRM) (**b**) to detect the mitochondrial respiration and membrane potential by flow cytometry, respectively. **c** Immunoblot analysis of Cytochrome c release and apoptosis in BMDMs as indicated treatment. **d, e** THP-1-derived macrophages were treated with DMSO or cyclosporine A (50 μM, red plots) for 1 h followed by PFOS (150 μM, 6 h), poly (dA:dT) (2 μg/ml, 6 h) treatment or pre-treated with LPS (200 ng/ml, 3 h) followed ATP (5 mM, 6 h) treatment. IL-1β secretion in the supernatants of the indicated THP-1-derived macrophages were determined by ELISA (**d**). The caspase-1 activation (p10) and IL-1β maturation (p17) in the cell supernatants or pro-caspase-1 and pro-IL-1β in the cell lysates were detected by immunoblot (**e**). **f** Immunoblot analysis of the supernatants and cell extracts of BMDMs pre-treated with DMSO or cyclosporine A (50 μM) for 1 h. BMDMs were then treated with PFOS (150 μM, 6 h), poly (dA:dT) (2 μg/ml, 6 h) or pre-treated with LPS (200 ng/ml, 3 h) followed ATP (5 mM, 6 h) treatment. **g, h** Wild type (WT, black plots), *CYPD* knockout (KO#1: red plots; KO#2: blue plots) THP-1-derived macrophages were treated with PFOS (150 μM) for 6 h. Relative enrichment of mtDNA in AIM2-pulldown material were determined (**g**). IL-1β production in the supernatants were measured by ELISA (**g**). The maturation of IL-1β in the supernatants or pro- IL-1β and CypD in lysates were detected by immunoblot (**h**). **i, j** Immunoblot analysis of Cytochrome c from cytosolic extracts and mitochondria from indicated cells pre-treated with DMSO or cyclosporine A (50 μM) for 1 h, followed with PFOS treatment (150 μM, 6 h) (**i**). Relative enrichment of mtDNA in AIM2-pulldown material from the THP-1-derived macrophages (WT, black plots; *CYPD* KO#2: blue plots) were checked (**j**). In **a, b, d, g** and **j**, all error bars, mean values ± SEM, *P*-values were determined by unpaired two-tailed Student's *t* test of *n* = 3 independent biological experiments. For **c, e, f, h** and **i**, similar results are obtained for three independent biological experiments. Source data are provided as a Source data file.

treatment induces apoptosis. Since GSDME could be cleaved and activated by caspase-3[49–51], we next sought to determine the participation of GSDME in the PFOS-induced AIM2-independent cell death by knocking down *GSDME* in the *AIM2* KO THP-1-derived macrophages (Supplementary Fig. 11f). However, we did not observe remarkable change of cell death between *GSDME* KD cells and control cells upon PFOS treatment (Supplementary Fig. 11g), suggesting that caspase-3-mediated cleavage of GSDME did not play a major role in the PFOS-induced AIM2-independent cell death. Hence, we speculated that other caspase-3 substrates than GSDME may be involved in AIM2-independent cell death. To confirm this hypothesis, we knocked down *CASPASE-3* in *AIM2* KO THP-1-derived macrophages (Supplementary Fig. 11f), and found that the decrease of AIM2-independent cell death by *CASPASE-3* knockdown appeared more pronounced than that by *GSDME* knockdown (Supplementary Fig. 11g). Together these results indicate that PFOS induces caspase-3-mediated cell death and AIM2-dependent pyroptosis.

Given the observation that PFOS induces apoptosis and mitochondrial Cyt c loss, we next sought to determine the mechanism of PFOS-induced mtDNA release during apoptosis. Firstly, treatment of THP-1-derived macrophages and BMDMs with cyclosporine A (CsA) (a ligand of cyclophilin D (CypD), a critical component of the mitochondrial permeability transition pore) apparently inhibited PFOS-induced caspase-1 activation and IL-1β maturation/secretion (Fig. 4d–f). Secondly, *CYPD* KO THP-1-derived macrophages showed partly but significantly decreased mtDNA release, IL-1β in response to PFOS treatment, indicating that CypD is involved in PFOS-induced mtDNA release (Fig. 4g, h). Furthermore, CsA could also impair the PFOS-induced release of Cyt c and mtDNA in the *CYPD* KO macrophages (Fig. 4i, j), suggesting that in addition to its effect on CypD, CsA might also suppress PFOS-induced Cyt c and mtDNA release via additional mechanism[52–54].

**PFOS-triggered BAX/BAK activation facilitates mtDNA accumulation**. We next determined whether PFOS-induced mtDNA releases through mitochondrial outer membrane permeabilization (MOMP) pathway. Previous studies showed that BAX and BAK oligomerization on the mitochondrial outer membrane can cause MOMP during apoptosis[46,55]. Therefore, we investigated whether PFOS could induce BAX and BAK form oligomers. We found that PFOS exposure reduced the mRNA and protein level of anti-apoptotic protein BCL-2, and enhanced the expression of pro-apoptotic proteins BAX/BAK, to induce BAX/BAK oligomerization in THP-1-derived macrophages (Fig. 5a–c). Several studies

showed that BAX/BAK oligomerization form macropores on the mitochondrial outer membrane that contributes to mtDNA release into cytosol[46,55]. Hence, to further assess the contribution of BAX and BAK in PFOS-induced mtDNA release, we generated *BAX/BAK* single and double knockout cells (Fig. 5d). The cell death, mtDNA release, and IL-1β production/maturation were significantly decreased by knockout of either *BAX* or *BAK*, suggesting that both BAX and BAK contributed to the MOMP (Fig. 5e–h). Moreover, the single *BAX* and *BAK* deficiency showed more mtDNA release, cell death and IL-1β secretion/maturation than *BAX/BAK* double knockout, suggesting that BAX/BAK play important roles in PFOS-induced inflammasome activation (Fig. 5e–h). Herein, we sought to investigate the underlying mechanism of PFOS-induced BAX activation. It was reported that the activated JNK can promote BAX translocation to mitochondrial through phosphorylation of 14-3-3 proteins[56]. We found that PFOS induced the phosphorylation and activation of JNK, then leading to BAX translocation (Supplementary Fig. 12a). The JNK inhibitor SP600125 could suppress the phosphorylation and activation of the JNK, BAX translocation, and finally leading to the decrease of IL-1β release under PFOS stimulation (Supplementary Fig. 12a-b). The increase of cytosolic free $Ca^{2+}$ triggers the activation of PKC, leading to the activation of JNK signaling[30–32]. We further explored the involvement of $Ca^{2+}$ and PKC in PFOS-induced JNK activation, and found that the phosphorylation of JNK was evidently inhibited in presence of $Ca^{2+}$ chelator or PKC inhibitor under PFOS treatment (Supplementary Fig. 12c). Therefore, these findings indicate that PFOS exposure could induce $Ca^{2+}$–PKC–JNK–BAX activation pathway. In addition, it has been reported that the BAX which translocates toward mitochondria, can interact with CypD[57–59], and we found that PFOS exposure could trigger the interaction between BAX and CypD (Supplementary Fig. 12d). Taken together, our data illustrate that BAX/BAK- and BAX/CypD-mediated pathway contributes to PFOS-triggered the mtDNA release, resulting in AIM2 inflammasome activation.

**PFOS-induced tissue inflammation is dependent on AIM2 inflammasome activation**. As shown in Supplementary Figs. 1 and 2, PFOS induced severe inflammation in multiple organs. Thus, we sought to explore whether the AIM2 inflammasome was involved in PFOS-induced tissue inflammation in vivo. We found that $Aim2^{-/-}$ mice had reduced liver, lung, and kidney inflammation and damage compared to WT mice and $Nlrp3^{-/-}$ mice by histological analysis (Fig. 6a). Consistent with the observations in $Aim2^{-/-}$ BMDMs, IL-1β secretion from the serums, peritoneal fluids, livers, lungs, and kidneys following PFOS challenge was

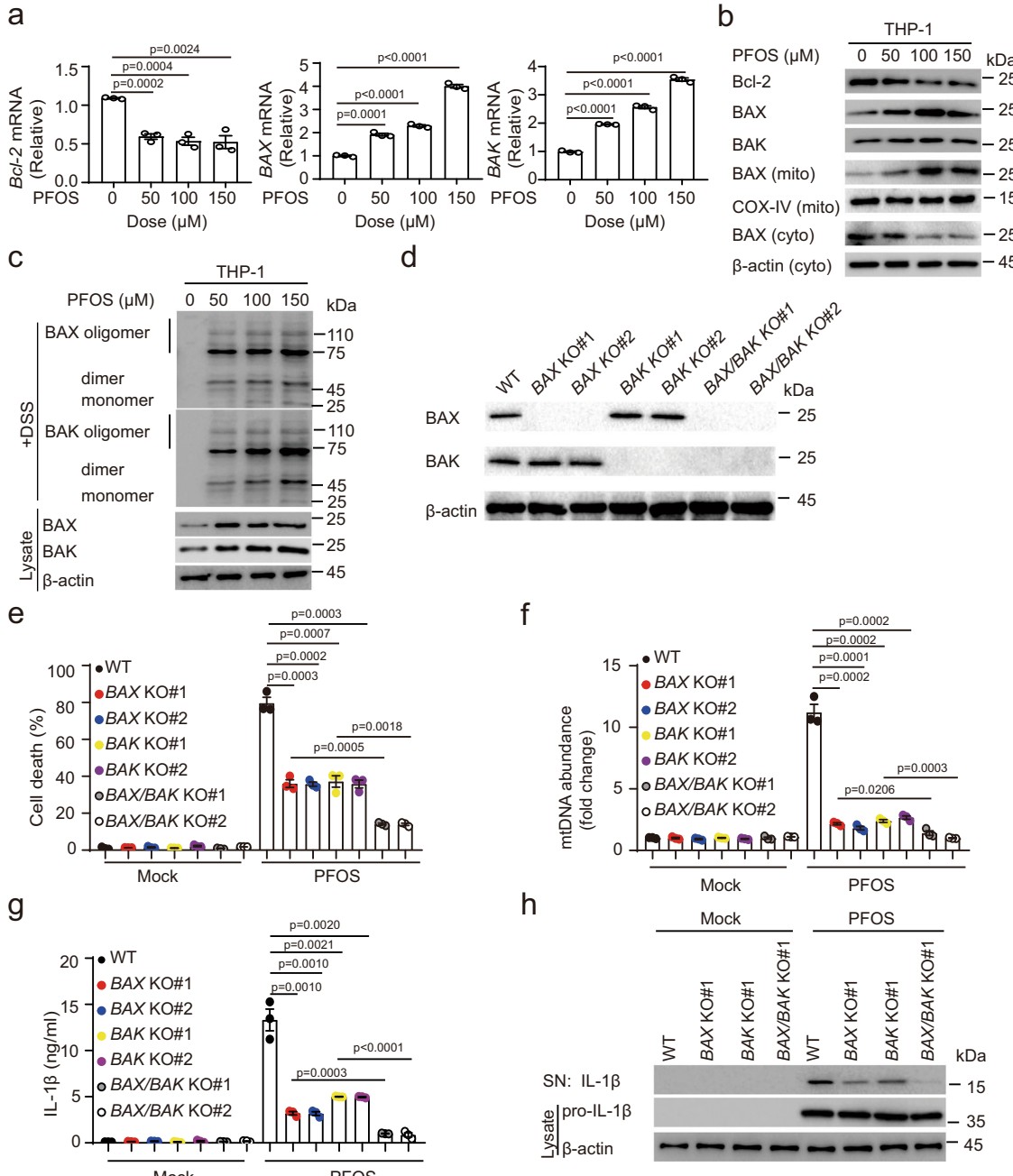

**Fig. 5 PFOS-induced BAX/BAK oligomerization contributes to mtDNA release. a** THP-1-derived macrophages were treated with PFOS as indicated for 6 h, cell lysates were collected to detect the mRNA levels of *BCL-2*, *BAX*, and *BAK*. **b** Immunoblot analysis of mitochondrial apoptosis pathway and BAX translocation in THP-1-derived macrophages treated as indicated. **c** Cell lysates and crosslinked pellets from THP-1-derived macrophages treated as indicated were analyzed by immunoblotting for BAX/BAK oligomerization. **d–h** Wild type (WT, black plots), *BAX* knockout (KO#1: red plots; KO#2: blue plots), *BAK* KO (KO#1: yellow plots; KO#2: purple plots) and *BAX/BAK* double KO (KO#1: gray plots; KO#2: white plots) THP-1-derived macrophages were generated (**d**). The indicated cells were treated with PFOS (150 μM) for 6 h. The cell death was determined by detecting the LDH release in the supernatants (**e**). Relative enrichment of mtDNA in AIM2-pulldown material was determined (**f**). IL-1β production in the supernatants was measured by ELISA (**g**). The maturation of IL-1β in the supernatants and pro-IL-1β in lysates were detected by immunoblot (**h**). In **a** and **e–g**, all error bars, mean values ± SEM, *P*-values were determined by unpaired two-tailed Student's *t* test of *n* = 3 independent biological experiments. For **b**, **c**, **d** and **h**, similar results are obtained for three independent biological experiments. Source data are provided as a Source data file.

remarkably reduced in *Aim2*$^{-/-}$ mice when compared to WT mice or *Nlrp3*$^{-/-}$ mice (Fig. 6b). However, compared with WT mice, the production of inflammasome-independent cytokine TNF-α and IL-6 were not altered in *Aim2*$^{-/-}$ mice (Fig. 6c, d). To further confirm whether AIM2-dependent IL-1β is responsible for the PFOS-induced tissue inflammation, the *Il-1β*$^{-/-}$ mice

were i.p. injected with PFOS. Consistently, the *Il-1β* deficiency protected the mice from tissue inflammation and damage under PFOS treatment (Supplementary Fig. 13a). In addition, to determine the cells compartment responsible for PFOS-triggered tissue inflammation, we next generated WT mice and *Aim2*$^{-/-}$ chimeric mice with adoptive bone marrow transplantation.

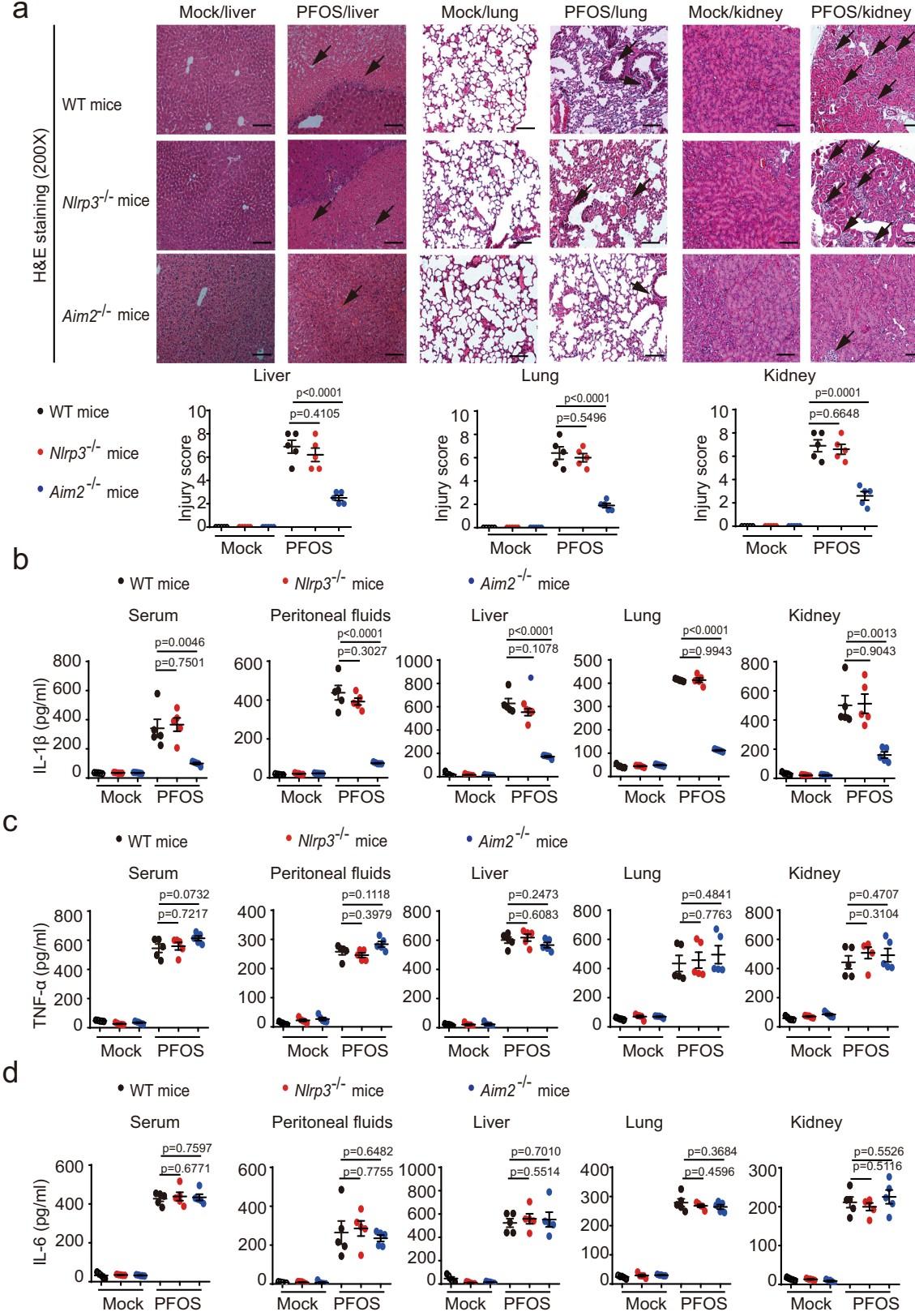

Among bone marrow recipients that were treated with PFOS, WT mice and *Aim2*−/− mice receiving WT mice bone marrow (referred to as WT > WT and WT > KO) showed similar tissue damage (Supplementary Fig. 13b). WT mice receiving *Aim2* KO bone marrow (referred to as KO > WT) had reduced liver, lung, and kidney inflammation and damage, compared to the

WT > WT group by histological analysis (Supplementary Fig. 13b). In addition, *Aim2* KO mice receiving *Aim2* KO bone marrow (referred to as KO > KO) had reduced liver, lung and kidney inflammation and damage, compared to the WT > KO group by histological analysis (Supplementary Fig. 13b). Interestingly, we found that the KO > WT group still had more severe

**Fig. 6 The effects of AIM2 inflammasome on PFOS-induced inflammatory responses in vivo. a–d** Wild type (WT), $Nlrp3^{-/-}$ (red plots) or $Aim2^{-/-}$ (blue plots) mice (female, 6 weeks old) were i.p. with PBS containing 2% Tween-80 (Mock, $n = 5$) or PFOS (dissolved in PBS containing 2% Tween-80, $n = 5$). In order to better show the protective role of inflammasome component deficiency in the presence of PFOS exposure, the dose of PFOS in this model we used was 25 mg/kg body weight per day. At 5 days post-treatment, liver tissue, lung tissue, and kidney tissue of these mice were stained with hematoxylin-eosin (H&E) and assayed using a light microscope with ×200 magnification. Scale bar, 100 μm. The tissue (liver, lung, and kidney) injury score was determined and averaged in 5 randomly selected nonoverlapping fields from respective individual mouse tissue sections. All histology analyses were conducted in a blinded manner (**a**). At 5 days post-treatment, the serum and peritoneal fluids were collected to determine the level of IL-1β (**b**), TNF-α (**c**), and IL-6 (**d**) by ELISA. The liver, lung, and kidney of treated mice were isolated and cultured for 24 h, and the secretion of IL-1β (**b**), TNF-α (**c**), and IL-6 (**d**) in the supernatants were detected by ELISA. In **b–d**, all error bars, mean values ± SD, P-values were determined by unpaired two-tailed Student's t test ($n = 5$ independent biological mice per group). Source data are provided as a Source data file.

symptoms of tissue inflammation than KO > KO (Supplementary Fig. 13b), indicating that despite its major role in the bone marrow-derived cells, AIM2 might also function in other cell types, which contributes to the subordinate PFOS-induced tissue inflammation and damage. Collectively, these observations highlight a vital role for AIM2 inflammasome activation in the development of PFOS-induced tissue inflammation and damage.

**PFOS induces AIM2 inflammasome activation to aggravate asthmatic inflammation.** Exposure of PFOS is associated with risk of asthma prevalence[7,60,61]; both animal model and in vitro cell model have indicated PFOS is involved in exacerbation of allergic asthma[10,62]. Herein, we further investigated if AIM2 inflammasome could mediate the PFOS-induced asthmatic exacerbation. We established an OVA-induced asthmatic mice model with continued exposure to PFOS to evaluate the additional effect of PFOS in asthma (Fig. 7a). As shown in WT mice, PFOS administration could promote the inflammatory cell infiltration and PAS-positive goblet cells in mice treated with PFOS + OVA as compared to OVA alone challenged mice (Fig. 7b, c). In $Aim2^{-/-}$ mice, the inflammatory patterns under PFOS + OVA treatment were markedly milder (showing a reduction of inflammatory cells and mucus production) than those from WT mice (Fig. 7b, c). In addition, PFOS further enhanced secretion of T helper cell 2 (Th2)-produced cytokine IL-4 and proinflammatory cytokine IL-1β in bronchoalveolar lavage fluid (BALF) and serum from asthmatic mice compared to OVA-only group; while the IL-4 and IL-1β levels were decreased in $Aim2^{-/-}$ mice exposed to PFOS + OVA compared with their WT phenotypes (Fig. 7d–f). Collectively, these observations indicated that PFOS-induced asthmatic exacerbation was also partially dependent on AIM2 inflammasome activation.

## Discussion

A growing number of pathogen-associated molecules and host-derived molecules that alert the immune system to cause cell injury have been identified as key triggers for inflammasome activation[63]. However, only a few studies have investigated the association of inflammasome and environmental stimulants (such as silica, particulate matters, and nanoparticles), and they have mainly focused on NLRP3 inflammasome[20–25]. Until now, the host sensing mechanisms in response to organic toxins have not been reported yet. Our study demonstrated that AIM2 activation is required for PFOS-induced inflammatory response and tissue damage, in which the process is dependent on mitochondrial dysfunction and release of mtDNA. The findings suggest an innate immune mechanism of host cells to recognize environmental organic pollutants.

When accumulation of PFAS in the tissues (like lung, liver, and kidney), they are engulfed by resident macrophages. This process results in the production of multiple proinflammatory cytokines (such as IL-1β, TNF-α, and IL-6), leading to cellular injury and

tissue inflammation[2]. Although PFOS-induced inflammatory response route has been reported, the molecular mechanism governing PFOS-induced inflammation has yet to be clarified. We reveal that PFOS disrupts mitochondrial function and promote the release of mtDNA but not genomic DNA into the cytoplasm, which consequently triggers AIM2 inflammasome activation. We further used a functional animal model to confirm that deficiency of the AIM2 inflammasome protected mice from PFOS-induced tissue inflammation or asthmatic exacerbation. Surprisingly, we did not find the ROS production in macrophages; and this seems to be contradicted to some previous studies, showing PFOS-induced oxidative damage via mitochondria-dependent pathway[14,19]. The disagreement is attributed that the injury in hepatocytes may be a downstream effect of PFOS-induced macrophage activation, i.e., PFOS induces inflammatory cytokines or mediators released by macrophages and in turn to cause oxidative stress in the structural cells in liver tissue. Nonetheless, our results elucidate the essential innate mechanisms of the host to sense PFOSs, suggesting that AIM2 could be an effective therapeutic target for PFOS-associated inflammatory diseases.

Damaged mitochondria release a variety of DAMPs (e.g., mtDNA and mtROS) into the cytosol to trigger oxidant injury, innate immune response, and cell death. Several studies have shown that oxidized mtDNA can act as a stimulant for NLRP3[39,40]. AIM2 is a well-described sensor to recognize either exogenous or endogenous double-stranded DNA[37,38,64]; while two recent reports have indicated the involvement of AIM2 inflammasome with mtDNA: Bae et al. demonstrated that circulating cell-free mtDNA in the serum from type 2 diabetic subjects induced AIM2 inflammasome activation in macrophages and contributed to the chronic inflammation in type 2 diabetes[65]; another report showed that cholesterol could induce mtDNA release and activation of AIM2 inflammasome in macrophages[66]. The above findings suggest that oxidized mtDNA specifically triggers NLRP3 activation, and non-oxidized mtDNA preferentially activated AIM2. Interestingly, our results didn't observe the production of mtROS in macrophages following PFOS stimulation, indicating limited oxidized mtDNA production in mitochondria; and this may be one reason why NLRP3 inflammasome was not activated by PFOS. Rather than NLRP3, the current results showed that PFOS could induce non-oxidized mtDNA release and trigger AIM2-dependent IL-1β secretion and pyroptosis. In our study, the presence of CsA significantly impaired Cyt c and mtDNA release into cytosolic compartment in PFOS-treated WT macrophages and $CYPD$ KO macrophages, indicating that CypD-activated pathway and additional CsA-sensitive mechanisms are involved in PFOS-induced mitochondrial contents release. Additionally, we found that BAX interacted with CypD and BAX/BAK formed oligomers in PFOS-exposed macrophages. Furthermore, either single KO of $BAX$ or $BAK$ could partly decrease PFOS-induced mtDNA release, suggesting that BAX or BAK alone has the ability to trigger macropores formation, which was similar with previous reports[67,68].

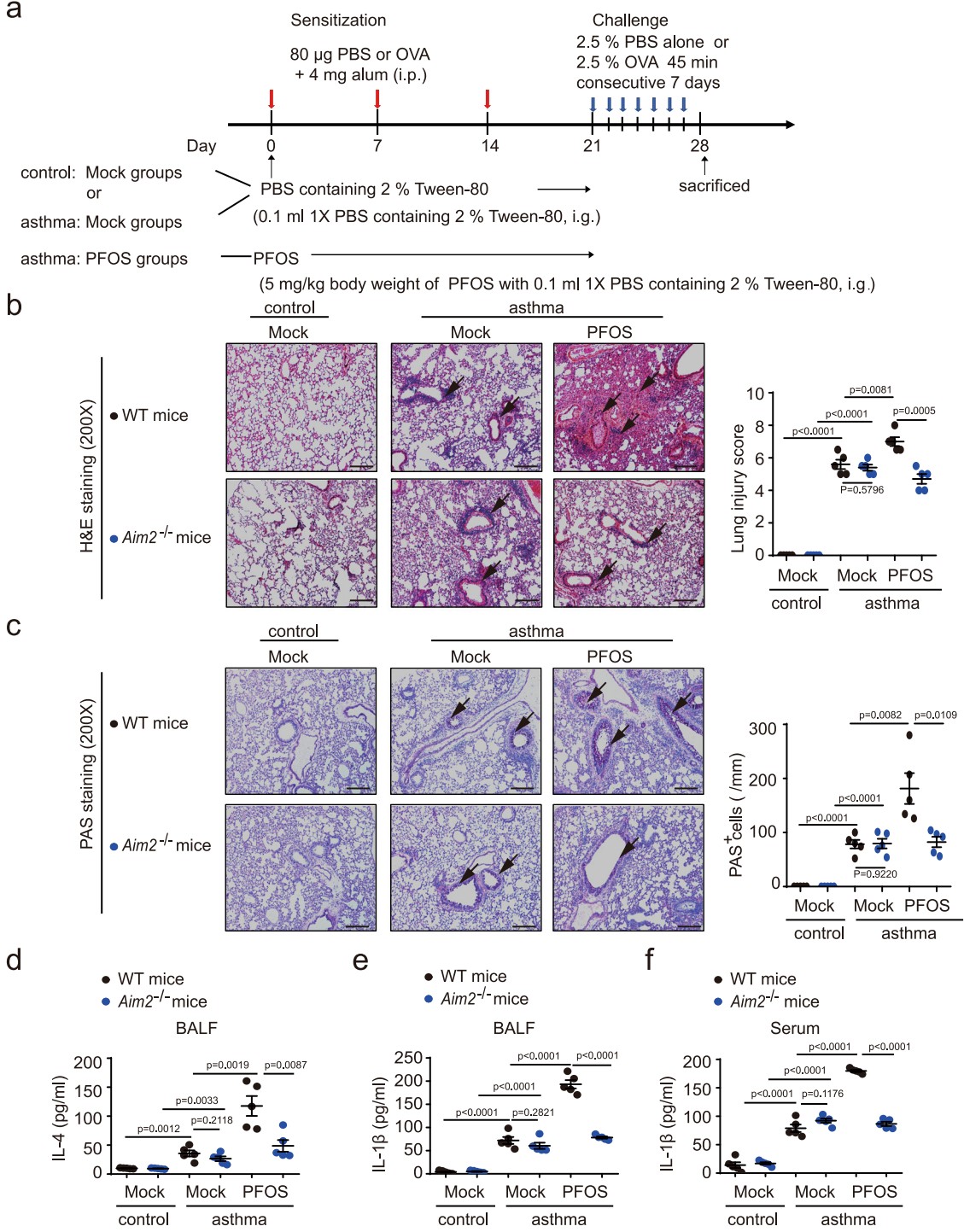

**Fig. 7 AIM2 inflammasome plays a critical role in PFOS-induced allergic asthma exacerbation. a** The model of PFOS-induced allergic asthma exacerbation. **b**–**f** Wild type (WT, black plots) or *Aim2*⁻/⁻ (blue plots) mice undergone intragastric administration with PBS or PFOS followed by OVA challenge, were sacrificed 24 h after the final OVA challenge. H&E staining (**b**) and periodic acid-Schiff (PAS) staining (**c**) of lung tissue were assayed using a light microscope with ×200 magnification. BALF or serum was collected 24 h after the final OVA challenge and ELISA analysis of IL-4 (**d**) in the BALF and IL-1β in the BALF (**e**) or serum (**f**). Scale bar, 100 μm. The lung injury scores and the number of PAS-positive cells per unit of length (mm) of the basement membrane were determined and averaged in five randomly selected nonoverlapping fields from respective individual mouse tissue sections. All histology analyses were conducted in a blinded manner. In **d**–**f**, all error bars, mean values ± SD, *P*-values were determined by unpaired two-tailed Student's *t* test (*n* = 5 independent biological mice per group). Source data are provided as a Source data file.

Furthermore, double KO of *BAX* and *BAK* almost eliminated PFOS-triggered mtDNA release. Thus, it is possible that mtDNA and Cyt c released into cytosolic compartment may be attributed to the mechanisms that BAX/BAK mediate MOMP in a CypD-dependent manner. It remains an interesting question that whether this process also involves mitochondrial permeability transition pores (MPTP) formation. A lot of evidence supported that BAX acts as a pivotal role in regulating MPT and MOMP via

interacting with CypD and BAK, respectively[47,52–55,57–59], the crosstalk between MPTP and MOMP under PFOS treatment deserves further study.

Previous studies indicated that various stimuli could induce inflammatory responses through the action of calcium-dependent mechanisms[69,70]. Generally, $Ca^{2+}$ signaling can be triggered by the release of $Ca^{2+}$ from intracellular ER storage, and the increase of cytosolic free $Ca^{2+}$ contributes to the activation of downstream kinases, such PKC, leading to the activation of NF-κB signaling or JNK signaling[30–34]. In our study, we showed that the protein level of ER stress monitor Bip and the level of cytosolic $Ca^{2+}$ were obviously upregulated after PFOS stimulation in BMDMs and THP-1-derived macrophages. PFOS has been reported to elevate the concentrations of cytosolic $Ca^{2+}$ in several cell types through L-type voltage-gated calcium channel[71,72]. Therefore, we assumed that PFOS might trigger calcium flux in human and mouse macrophages through L-type voltage-gated calcium channel which needs to be further clarified. Additionally, the application with $Ca^{2+}$ chelator BAPTA-AM or PKC inhibitor chelerythrine chloride could significantly suppress the PFOS-induced NF-κB and JNK signaling pathway activation. Furthermore, the JNK inhibitor SP600125 could apparently inhibit BAX translocation to mitochondria, supporting the notion that activated JNK contributed to BAX translocation. Collectively, these results suggest that $Ca^{2+}$-dependent PKC pathway plays a critical role in activating JNK-BAX and NF-κB signaling pathway under PFOS stimulation in macrophages.

Although many inflammasome activators have been identified, it remains unclear how these diverse stimuli trigger inflammation activation. One model points out that the NLRs directly interact with the stimulants; another model suggests that these divergent stimuli are recognized through indirect pathways (via induction of endogenous "danger" molecules). With regard to the diversity of environmental factors, how the phagocytosed pollutants are sensed by NLRs needs to be further clarified. Because the environmental pollutant itself doesn't have biological "components" (like DNA/RNA, proteins, and peptides), it seems unlikely these stimulants can be directly recognized by host sensors. Recognition of silica by NLRP3 is a good example to explain that inflammasome is triggered by silica-induced lysosomal damage and rupture but not silica itself[21]. Our data support this model in which non-oxidized mtDNA induced by PFOS can trigger AIM2 inflammasome activation. Given the mechanisms discussed above, it is possible that the AIM2 or other NLRs are activated through some common endogenous danger molecules downstream of diverse organic stimuli. It would be of interest to screen more DAMPs induced by different organic chemicals and to determine the key host sensors responding to these stimuli.

In the current study, we tried to mimic the human exposure level of PFOS by using the mouse model. Based on the results from the epidemiological studies, the serum level of PFOS in general population ranges from 0.2 to 99.7 ng/ml[35,36], while PFOS level can reach up to 92,303 ng/ml in those people who are highly exposed to perfluoroalkyl substances (such as the fishery employees and fluorochemical production workers)[73,74]. We have found that serum PFOS level in the same C57BL/6 mice which were exposed to PFOS at dosage of 7 mg/kg/day via gavage for 28 days (total exposure amount 196 mg/kg) was about 8434 ng/ml, i.e., every 1 mg/kg PFOS exposure in mouse is equivalent to 43 ng/ml (8430/196) serum PFOS concentration[75]. In the current study, we can speculate that the serum PFOS levels in acute exposure model (5, 15, and 25 mg/kg for 5 days) are about 1075, 3225, and 5375 ng/ml, respectively; while serum PFOS levels in chronic exposure model (0.066 mg/kg for 30 days; 5 mg/kg for 20 days) is about 86 ng/ml and 4300 ng/ml, respectively. In addition, both acute and chronic exposure model demonstrate significant damage effects on multiple organs. Hence in our experimental model, the PFOS administration

dosages and expected PFOS internal exposure levels should be comparable to the occupational exposure levels and normal exposure levels in human.

Regarding the complexities of PFOS-induced inflammatory response, other forms of mitochondrial damage, apoptosis-associated cellular injury, and the other inflammation signaling cascades (such as NF-κB) merit further investigation. In addition, how innate immune response leads to the chronic inflammatory conditions in multiple organs exposed to PFOS also needs to be clarified. Nonetheless, our findings reveal that $Ca^{2+}$-PKC-NF-κB/JNK-BAX/BAK-mtDNA-AIM2 axis is a critical innate mechanism for PFOS-induced inflammation (Fig. 8), and may have important implications for the development of therapy against PFOS-induced toxicity and tissue damage.

## Methods

**Animals.** C57BL/6J $Aim2^{-/-}$ mice were kindly provided by Dr. Bin Sun, Shanghai Institute of Biochemistry and Cell Biology[76]. C57BL/6J $Nlrp3^{-/-}$ mice and C57BL/6J $Il-1\beta^{-/-}$ mice were kindly provided by Dr. Rongbin Zhou, University of Science and Technology of China[77,78]. C57BL/6J WT mice (GDMLAC-O7) were purchased from Guangzhou Medical Laboratory Animal Center of China. Animals were kept and bred in a specific-pathogen free (SPF) environment with standard conditions of temperature (20–26 °C) and humidity (40–70%) under a strict 12 h light cycle (lights on at 08:00 a.m. and off 08:00 p.m.) at Sun Yat-sen University, approved all the experimental protocols concerning the handing of mice. All animal experiments protocols were approved by the Animal Care Committee of the Sun Yat-sen University (Authorization number: SYXK (YUE) 2017-0175, Guangzhou, China). The mice were euthanized by $CO_2$ from compressed gas cylinders, and we complied with all the ethical regulation.

**PFOS-induced tissue inflammation mouse model.** WT, $Nlrp3^{-/-}$, $Aim2^{-/-}$, or $Il-1\beta^{-/-}$ mice (female; 6 weeks old) were intraperitoneally injected (i.p.) with PFOS dissolved in phosphate-buffered saline (PBS, Corning) containing 2 % Tween-80 (Vetec) for experiment groups ($n = 5$; acute exposure: 5, 15, and 25 mg/kg body weight per day for 5 days; chronic exposure: 0.066 mg/kg body weight per day for 30 days) or i.p. with PBS containing 2% Tween-80 (control groups, $n = 5$) for 5 or 30 days. The PFOS intraperitoneal injection model was modified by previous PFOS animal studies[27–29]. The experiment groups and control groups were maintained and bred separately in the same specific-pathogen-free environment at Sun Yat-sen University. At 5 days or 30 days post-treatment, mice liver, lung and kidney tissue were isolated and cultured for 24 h, and supernatants were analyzed by ELISA. Peritoneal cavities were washed with 3 ml of PBS. The peritoneal fluids and serum were harvested and concentrated for ELISA analysis.

**Isolation and culture of the tissue.** The mice were euthanized by $CO_2$ from compress gas cylinders and then fully soaked them into 75% ethanol (Guangzhou Chemical Reagent Factory, GCRF, China) for 15 s to reduce the possibility of infection. Dissect and remove the entire liver, lung, and kidney into a petri dish containing 1% fetal bovine serum (FBS, Gibco) of sterile PBS, respectively. Cut out the gallbladder from the liver and wash the tissue using 1% FBS of sterile PBS for three times to ensure the tissue clear of blood. Cut the liver, lung, kidney into 1 mm³, and digested by the medium A (sterile PBS, 1 M HEPES (Gibco), 5% KCL (GCRF, China), 1 M glucose (Sigma-Aldrich), 500 mM $CaCl_2$ (GCRF, China), phenol red solution (Sigma-Aldrich). pH adjusted to 7.4 with 2 M NaOH) containing 0.2% IV-Collagenase (Sigma) in 37 °C for 15 min, respectively. Stop the digestion with DMEM medium (Hyclone) with 10% (vol/vol) FBS. Filter the crude cell suspensions through gauze mesh filter (100 μm in diameter, FALCON) and transfer the resulting cell suspensions into sterile tubes and centrifuge at $60 \times g$ for 5 min. Discard the supernatant and repeat the wash procedure for three times via using medium A to resuspend the cell pellets and centrifuging at $60 \times g$ for 3 min. Resuspend the cells in cultured medium (DMEM medium containing 10% FBS and 1% L-glutamine) and check cells numbers and viability using hemocytometer and trypan blue. The cells are cultured into 60 mm tissue culture dishes (Thermo Fisher Scientific) via adding 5 ml of cell suspension with $1 \times 10^7$ cells per ml and softly rock the dish to ensure uniform distribution of cells. After culturing them at 37 °C with 5% $CO_2$ in an incubator for 24 h, cell supernatants are collected to determine the production of proinflammatory cytokines (IL-1β, TNF-α, and IL-6) by ELISA.

**Bone-marrow transplant.** For performing bone marrow transplant, the recipient mice (6 weeks old) were given antibiotic (Gentamicin,1 mg/mL, Sangon, China) in drinking water for 2 days before given 900 cGy irradiation. Donor bone marrow cells ($5 \times 10^6$) were obtained from the donor mice (6 weeks old) were intravenous injected into the recipient mice (after irradiation) and followed with antibiotic drinking water for 1 week. The recipient mice and donor mice were maintained and housed separately in the same specific-pathogen-free environment at Sun Yat-sen University.

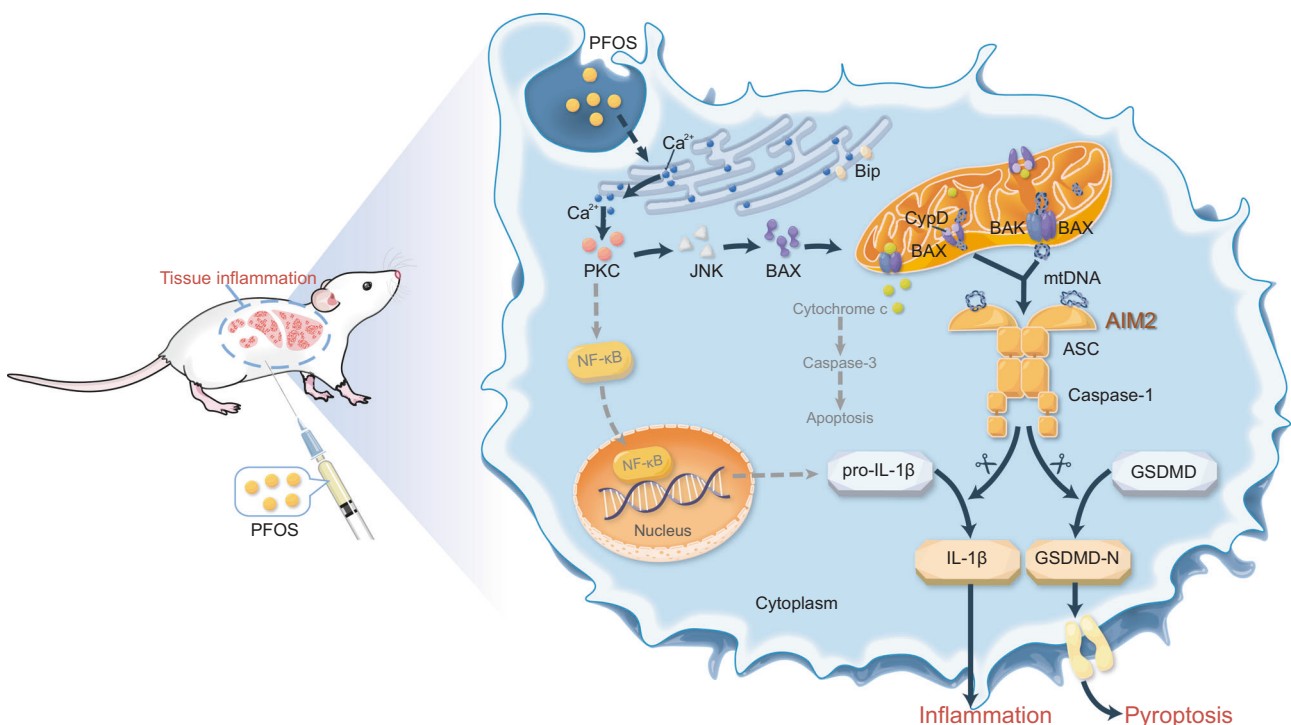

**Fig. 8 Schematic diagram to illustrate the mechanisms that PFOS exposure induces AIM2 inflammasome activation to promote tissue damage.** PFOS induces mitochondrial DNA (mtDNA) release through the Ca²⁺-PKC-NF-κB/JNK-BAX/BAK axis; and PFOS-induced mtDNA serves as an essential danger-associated molecular pattern (DAMP) to trigger AIM2 inflammation activation, ultimately leading to tissue inflammation.

**PFOS-induced asthmatic exacerbation mouse model**. The PFOS exposed allergic asthmatic mouse model was established based on the previous report[11]. Briefly, mice were exposed to PFOS or PBS containing 2% Tween 80, via gavage from day 0 to day 20. The ovalbumin (OVA) and OVA combined PFOS groups were first sensitized with OVA and alum by subcutaneous injection on days 0, 7, and 14; and then challenged with an aerosol of 2.5% OVA using an ultrasonic nebulizer on day 21–27. WT mice (female, 6 weeks old) or $Aim2^{-/-}$ mice (female, 6 weeks old) were administrated with PBS containing 2% Tween 80 (mock group, $n = 5$), OVA (asthmatic group, $n = 5$), and OVA plus PFOS (asthmatic exacerbation group, $n = 5$). The mice for mock group (control group), asthmatic group, and asthmatic exacerbation group were kept and housed separately at Sun Yat-sen University with the same specific-pathogen-free environment. On day 28, all mice were sacrificed by $CO_2$ from compress gas cylinders and assessed. The serums were collected for cytokine analysis; the lung tissues were collected for H&E staining and PAS staining. The bronchoalveolar lavage fluid (BALF) was collected as described previously[78]. Briefly, the lungs of sacrificed mice were injected with 1 ml PBS containing 5 mM EDTA through the trachea and then a lavage step was followed. The BALF samples were then centrifuged at $350 \times g/4\,°C$ for 5 min and the supernatants were harvested for cytokines detection.

**Cells**. THP-1 cells obtained from the Cell Bank of the Chinese Academy of Sciences (Shanghai, China) were maintained in RPMI 1640 medium (Gibco) with 10% (vol/vol) fetal bovine serum and 1% L-glutamine (Gibco) at 37 °C with 5% $CO_2$ in an incubator. THP-1 cells were differentiated into macrophages cultured with RPMI 1640 containing 100 ng/ml phorbol-12-myristate-13-acetate (PMA) for 12 h, and then the macrophages had a resting period containing 24 h before stimulated. Bone marrow cells (BMs) were isolated from randomly chosen 6 weeks old of wild-type or $Nlrp3^{-/-}$ or $Aim2^{-/-}$ mice and then BMs were differentiated into bone marrow-derived macrophages (BMDMs) cultured in 10 ml conditioned medium (DMEM medium (Corning) with 10% fetal bovine serum, 1% L-glutamine, 1% penicillin–streptomycin (Gibco), 100 ng/ml macrophage colony-stimulating factor (PeproTech) at 37 °C in an atmosphere containing 5% $CO_2$ in an incubator. On day 3, most cells were adherent to the dish. Discard the medium from the dish, and then the dish was supplemented with 10 ml of conditioned medium for further growth until day 6. On day 7, the cells were used to perform the experiments.

**Reagents and antibodies**. LPS (cat# L4524, for priming), PFOS (cat# 77282), EtBr (cat# E1510), puromycin (cat# P9620), albumin from chicken egg white (OVA, cat# 15503), PMA (cat# P1585), pyruvate (cat# P2256), uridine (cat# u3750), anti-Flag (M2, cat# F1804) and anti-β-actin (cat# A1978) were purchased from Sigma-Aldrich. ATP (cat# tlrl-atpl), poly(dA:dT) (cat# tlrl-patn-1), MCC950 (cat# inh-mcc) were purchased from Invivogen. Mito-TEMPO (cat# ALX-430-150-M005)

were from Enzo Life Sciences. CsA (cyclosporine A) (cat# 59865-13-3), SP600125 (cat# HY-12041) and BAPTA-AM (cat# HY-00545) were from MCE. Imjet$^{Tm}$ Alum (cat# 77161) and DSS (disuccinimidyl suberate) (cat# 21655) were from Thermo Scientfc. Chelerythrine Chloride (cat# SC0317) and Fluo-4 AM (cat# S1060) were from Beyotime Technology. Protein A agarose (20333) and protein G agarose (20399) were from Pierce. Anti-Bip (cat# AB310) was from Beyotime Technology. Anti-AIM2 (cat# ab93015) and anti-COX-IV (cat# ab16056) was from Abcam. Anti-ASC (cat# sc-271054), anti-caspase-1 (cat# sc-515), anti-Bcl-2 (cat# sc-7382), anti-GSDMD (cat# sc-81868), anti-GSDME (cat# sc-393162), anti-Cyclophilin D/Cyclophilin F (cat# sc-376061), goat anti-mouse (cat# sc-2005) and goat anti-rabbit (cat# sc-2004) were purchased from Santa Cruz Biotechnology. Anti-JNK (cat# 9252), anti-phospho-SAPK/JNK (cat# 9255), anti-BAX (cat# 5023), anti-BAK (cat# 12105), anti-Cytochrome c (cat# 4272), anti-cleaved caspase-3 (cat# 9664), anti-cleaved PARP (cat# 5625), anti-IL-1β (cat# 12242) and anti-caspase-1 (cat# 2225,), anti-IκBα (cat# 4814), anti-p65 (cat# 6956) and anti-p-p65 (cat# 3033) were purchased from Cell Signaling Technology. Anti-NLRP3 (cat# A27391510) was purchased from AdipoGen. The dilutions of indicated antibodies are listed in Supplementary Table 1.

**ASC and BAX/BAK oligomerization detection**. THP-1-derived macrophages or BMDMs were seeded in 6-well plates and treated as indicated. The cell pellets were collected into 1.5 ml Eppendorf tubes and lysed in TBS buffer (50 mM Tris-HCl (GCRF, China), 150 mM NaCl (GCRF, China), 0.5% Triton X-100 (Sigma-Aldrich), PH 7.4) with phosphatase inhibitor (Roche) and EDTA-free protease inhibitor (Bimake) on a rocker for 30 min on ice, and then centrifuged at $6000 \times g/4\,°C$ for 15 min to discard the supernatants. The cell supernatants were harvested for precipitation and detected the activation of caspase-1 (p10) and the maturation of IL-1β (p17) by western blotting. The pellets were washed twice with TBS buffer and resuspend in TBS buffer containing 2 mM fresh disuccinimidyl suberate (DSS, Thermo Fisher Scientific) cross-linker at 37 °C for 30 min to crosslink with flipping the tubes every 10 min, and then spun at $6000 \times g/4\,°C$ for 15 min. The crosslinked pellets were resuspended in 25 µl 2× SDS loading buffer (Cell Signaling Technology) and boiled at 100 °C for 5 min, and analyzed by immunoblotting of anti-ASC antibody, anti-BAX antibody or anti-BAK antibody.

**Immunoblot analysis**. Relevant cells were stimulated as indicated, then collected in lysis buffer (50 mM Tris, pH 8.0, 150 mM NaCl, 1 mM EDTA (Vetec), 1 mM DTT (Vetec), and 0.5% IGEPAL CA630 (Sigma-Aldrich) supplemented with phosphatase inhibitor and EDTA- free protease inhibitor) and incubated on a rocker with ice for 30 min. Cell lysates were centrifuged at $12,000 \times g/4\,°C$ for 15 min and lysates were boiled at 100 °C for 5 min in 5× SDS Loading Buffer and resolved by SDS-PAGE. Proteins were transferred into polyvinylidene fluoride (PVDF)

membrane (Bio-Rad) and then incubated with the appropriate antibodies. Immobilon Western Chemiluminescent HRP Substrate (Millipore) was used for protein detection. Images were performed using the ChemiDoc MP System (Bio-Rad) and Image LAb version 6.0 (Bio-Rad software, California, USA).

**Generation of CRISPR/Cas9 knockout cell lines.** After THP-1 cells were seeded, medium was replaced with DMEM containing lentiviruses expressing Cas9 and specific sgRNAs, added polybrene (8 µg/ml) (Sigma-Aldrich) for 48 h. Cells were selected for puromycin (Sigma-Aldrich) resistance, and polyclonal pools of cells were used for clonal screening and following experiments. The small-guide RNA (sgRNA) sequences targeting indicated genes were obtained from Sangon (Shanghai, China) listed follows:

*CASPASE-1* guide RNA: #1, 5′-TCCACTAGCATCTTACCTTG-3′
#2, 5′-CCGACTTTTGTTTCCATATCCTT-3′
*ASC* guide RNA: #1, 5′-CATGTCGCGCAGCACGTTAGCGG-3′
#2, 5′-GGTGCTGGTCTATAAAGTGCAGG-3′
*NLRP3* guide RNA: #1, 5′-AAGGAAGAAGACGTACACCG-3′
#2, 5′-CACCGTGGTGTTCCAGGGGGCGG-3′
*AIM2* guide RNA: #1, 5′-CCTTATCCTACCTTAACATG-3′
#2, 5′-TTCAGCATCTAACACACGTGAGG-3′
*CYPD* guide RNA: #1, 5′-GTGT TCGG TCAC GTCA AAGA-3′
#2, 5′-CCGG GAAC CCGC TCGT GTAC-3′
*BAX* guide RNA: #1, 5′-CCCCCCGAGAGGTCTTTTTCCGA-3′
#2, 5′-CGAGTGTCTCAAGCGCATCGGGG-3′
*BAK* guide RNA: #1, 5′-GCATGAAGTCGACCACGAAGCGG-3′
#2, 5′-GTTGATGTCCTCCCCGATGATGG-3′

**siRNA Transfection.** Chemically synthesized 21-nucleotide siRNA duplexes were obtained from Sangon (Shanghai, China) and transfected using Lipofectamine RNAiMAX (Invitrogen) according to the manufacturer's instructions. RNA oligonucleotides used are as follows:

Control siRNA, 5′-GUUAUCG CAACGUGUCACGUA-3′
Human *GSDME* siRNA #1, 5′-GCUUCUAAGUCUGGUGACAAATT-3′
Human *GSDME* siRNA #2, 5′-GCAUUCAUAGACAUGCCAGAUTT-3′
Human *CASPASE-3* siRNA #1, 5′-CCGACAAGCUUGAAUUUAU-3′
Human *CASPASE-3* siRNA #2, 5′-GGGAGACCUUCACAAACUUTT-3′

**RNA extraction and quantitative RT-PCR.** Relevant cells were treated as indicated and total RNA was extracted from cells using Trizol reagent (Invitrogen) following the manufacturer's protocols, then reverse-transcribed into cDNA using HiScript® II Q RT SuperMix (Vazyme). The cDNA was determined by quantitative RT-PCR using 2 × RealStar SYBR Mixture (Genestar) with the primers (listed in Supplementary Table 2) as follows:

Human IL-1β forward: 5′-AGCTACGAATCTCCGACCAC-3′
Human IL-1β reverse: 5′-CGTTATCCCATGTGTCGAAGAA-3′
Human *TNF-α* forward: 5′-GAGGCCAAGCCCTGGTATG-3′
Human *TNF-α* reverse: 5′-CGGGCCGATTGATCTCAGC-3′
Human *IL-6* forward: 5′-ACTCACCTCTTCAGAACGAATTG-3′
Human *IL-6* reverse: 5′-CCATCTTTGGAAGGTTCAGGTTG-3′
Human *BCL-2* forward: 5′-GGTGGGGTCATGTGTGTGG-3′
Human *BCL-2* reverse: 5′-CGGTTCAGGTACTCAGTCATCC-3′
Human *BAX* forward: 5′-CCGAGAGGTCTTTTTCCGAG-3′
Human *BAX* reverse: 5′-CCAGCCCATGATGGTTCTGAT-3′
Human *BAK* forward: 5′-GTTTTCCGCAGCTACGTTTTT-3′
Human *BAK* reverse: 5′-GCAGAGGTAAGGTGACCATCTC-3′
Human *GAPDH* forward: 5′-GGAGCGAGATCCCTCCAAAAT-3′
Human *GAPDH* reverse: 5′-GGCTGTTGTCATACTTCTCATGG-3′
Mouse *Il-1β* forward: 5′-GCAACTGTTCCTGAACTCAACT-3′
Mouse *Il-1β* reverse: 5′-ATCTTTTTGGGGTCCGTCAACT-3′
Mouse *Tnf-α* forward: 5′-GACGTGGAACTGGCAGAAGAG-3′
Mouse *Tnf-α* reverse: 5′-TTGGTGGTTTGTGAGTGTGAG-3′
Mouse *Il-6* forward: 5′-TAGTCCTTCCTACCCCAATTTCC-3′
Mouse *Il-6* reverse: 5′-TTGGTCCTTAGCCACTCCTTC-3′
Mouse *Gapdh* forward: 5′-AGGTCGGTGTGAACGGATTTG-3′
Mouse *Gapdh* reverse: 5′-TGTAGACCATGTAGTTGAGGTCA-3′

**Flow cytometry.** For mitochondria functional studies, relevant cells were stimulated as indicated and then stained with MitoSOX (2.5 µM, Life Technologies), Mitotracker Deep Red (100 nM, Thermo Fisher Scientific) or TMRM (100 nM, Thermo Fisher Scientific) at 37 °C for 30 min. Then the cells were washed three times with ice PBS and suspended in PBS for analyzing. The gating strategy for the detection of Mito-SOX-, Mitotracker Deep Red- and TMRM-negative and positive cells was showed in Supplementary Fig. 14a. For apoptotic studies, cells were treated as indicated and then cells were collected for detecting apoptosis using Annexin V-FITC/7AAD Kit (4A Biotech, Beijing, China, FXP018) according to the manufacturer's protocols. The gating strategy for apoptotic cells detection was showed in Supplementary Fig.14b. Above samples were analyzed by flow cytometry using a guava easyCyte HT system (Millipore) and FlowJo version 10.0 (Tree Star software, Ashland, USA).

**Cytosolic Ca$^{2+}$ detection.** The relevant cells were treated as indicated and then washed three times with ice PBS and incubated with Flo-4 AM (5 µM) at 37 °C for 30 min. Then the cells were washed three times with ice PBS and then suspended in PBS for incubating for another 20 min at 37 °C. The fluorescence was determined by using fluorescence microplate reader (excitation and emission wavelength at 488 nm and 520 nm, respectively). Cytosolic Ca$^{2+}$ content was presented as the percentage relative to the value of untreated cells.

**Measurement of cytokines.** Cytokines production was measured with human IL-1β kit (BD Biosciences, cat. 557953), TNF-α kit, BD Biosciences, cat. 555212) and IL-6 kit (BD Biosciences, cat. 555220) according to the manufacturer's protocols, respectively. Mouse IL-1β, TNF-α, IL-6, and IL-4 production in cell supernatants were detected with ELISA kits (BD Biosciences, 59603; BD Biosciences, 558534; BD Biosciences, 555240; Invitrogen, cat. 88-7044-22) according to the manufacturer's protocols, respectively.

**Cytotoxicity assay.** Cells were treated as indicated and then supernatants were collected for detecting cell death by a lactate dehydrogenase (LDH) assay using CytoTox 96 Non-Radioactive Cytotoxicity kit (Promega, cat. G1780) according to the manufacturer's protocols

$$\text{Cytotoxicity}\,(\%)\,\text{or Cell death}\,(\%) = [(\text{Experimental LDH release} - \text{Medium background})$$
$$/\,(\text{Maximum LDH release control} - \text{Medium background})] \times 100$$

**Generation of mitochondrial DNA deficient ($\rho^0$) cells.** THP-1 cells and BMDMs were grown in the mediums (RPIM 1640 medium for THP-1 cells, DMEM medium for BMDMs) containing 10% fetal bovine serum, 1% L-glutamine, pyruvate (100 µg/ml), uridine (50 µg/ml) and EtBr (100 ng/ml) for 7 days at 37 °C with 5 % CO$_2$ in an incubator. The total DNA of indicated cells were extracted by using QIAamp DNA Micro Kit (QIAGEN) according to the manufacturer's instruction. Mitochondrial DNA depletion was quantified by quantitative RT-PCR with mtDNA (*D-loop*) and nucDNA (*Tert*) primers (listed in Supplementary Table 2).

**Cytosolic and total DNA extraction.** $1 \times 10^7$ THP-1-derived macrophages were treated with PFOS as indicated, washed three times with ice cold PBS, and trypsinization off the dish. Cells were pelleted by spinning at $800 \times g/4$ °C for 5 min. Then we normalized the protein concentration and volume of the supernatant. One of the cell pellets was resuspended with cytosolic extraction buffer (150 mM NaCl, 50 mM HEPES pH 7.4, 25 µg/ml digitonin (MCE) and homogenized at 4 °C for 10 min, followed by centrifugation at $10,000 \times g/$ 4 °C for 15 min to get a supernatant for the cytosolic fraction. Another one was centrifuged to produce cell pellets to extract total DNA. The cell pellets and cytosolic fraction were used to extract total and cytosolic DNA with a QIAamp DNA Micro Kit (QIAGEN) according to the manufacturer's instruction, respectively. DNA was assessed using quantitative RT-PCR with mtDNA (*D-loop*) and nucDNA (*Tert*) primers as listed in Supplementary Table 2.

**Quantification of AIM2-bound mitochondrial DNA.** The protocol for quantification of AIM2-bound DNA was modified as shown in the previous report[79]. Briefly, THP-1-derived macrophages were treated with either mock or PFOS (150 µM) for 6 h, then the cell extracts were immunoprecipitated by incubation with anti-AIM2 plus Protein A/G beads (Pierce). Beads were washed with lysis buffer for two times, and the immunoprecipitates were eluted with TE buffer (Omega) then incubated at 65 °C for 12 h. DNA precipitation was carried out with isopropanol (GCRF, China) and evaluated by qRT-PCR. 10 ng of the EGFP (enhanced green fluorescent protein) plasmid (CLONTECH Laboratories, cat# 6083-1) was added into mock or PFOS condition. The mixture of EGFP plasmid and endogenous DNA was used to assay the presence of the DNA fragments by using the primers of human mitochondrial DNA. Relative levels of the DNA were normalized to Ct values of *EGFP* gene amplification. The primer sequences (listed in Supplementary Table 2) as follows:

*EGFP* forward: 5′-ACGGCGACGTAAACGGCCAC-3′
*EGFP* reverse: 5′-GCACGCCGTAGGCTAGGGTG-3′
*mtDNA1* forward: 5′-CACCCAAGAACAGGGTTTGT-3′
*mtDNA1* reverse: 5′-TGGCCATGGGTATGTTGTTAA-3′
*mtDNA2* forward: 5′-CTATCACCCTATTAACCACTCA-3′
*mtDNA2* reverse: 5′-TTCGCCTGTAATATTGAACGTA-3′
*mtDNA3* forward: 5′-AATCGAGTAGTACTCCCGATTG-3′
*mtDNA3* reverse: 5′-TTCTAGGACGATGGGCATGAAA-3′
*Myc* forward: 5′-AAGGACTATCCTGCTGCCAA-3′
*Myc* reverse: 5′-CCTCTTGACATTCTCCTCGG-3′
*Sox2* forward: 5′-TTTTGTCGGACGAGGAAAG-3′
*Sox2* reverse: 5′-CATGAGCGTCTTGGTTTTCC-3′
*18 S* forward: 5′-TAGAGGGACAAGTGGCGTTC-3′
*18S* reverse: 5′-CGCTGAGCCAGTCAGTGT-3′

**Histological assessment.** The mice were euthanized by CO$_2$ from compressed gas cylinders. Then the tissues were removed and fixed in 4% paraformaldehyde (Meilunbio, China) for more than 24 h, and embedded in paraffin. The sections (thickness, 6 µm) were stained with hematoxylin & eosin or Periodic acid-Schiff

(PAS) solutions (Servicebio, China). The tissues injury scores and the number of PAS-positive cells per unit of length (mm) of the basement membrane were determined in 5 randomly selected nonoverlapping fields from respective individual tissue sections using a microscope (DMi8; Leica) and all histology analyses for H&E staining were conducted in a blinded manner as a combination of tissue damage (score 0–5) and inflammatory cell infiltration (score 0–5) with a total ranging from 0 to 10. The number of PAS-positive cells per unit of length (mm) of the basement membrane were assessed with ImageJ software version 1.52 (National Institutes of Health software, Maryland, USA).

**Statistical analysis**. The data of all quantitative in vitro experiments are represented as mean ± SEM of three independent experiments; data of in vivo experiments are represented as mean ± SD of five mice per group. Unpaired two-tailed Student's t-test with a P-value <0.05 considered statistically significant was used for all statistical analyses using the GraphPad Prism version 8.0 (GraphPad Software, California, USA).

**Reporting summary**. Further information on research design is available in the Nature Research Reporting Summary linked to this article.

## Data availability
The data that support this study are available within the article and its Supplementary Information files or available from the authors upon request. Source data are provided with this paper.

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

## Acknowledgements

The authors thank Prof. Marius Sudol for editing the manuscript. This work was supported by the National Key R&D Program of China (2020YFA0908700), National Natural Science Foundation of China (92042303, 31870862, 31700760, 31800751, 81770983, and 81974139), Science and Technology Planning Project of Guangzhou, China (201907010038), Guangdong Basic and Applied Basic Research Foundation (2020B1515120090), and the Fundamental Research Funds for the Central Universities (18lgpy49 and 18lgpy53).

## Author contributions

L.W. performed the investigation and performed the analysis. T.L., S.Y., L.S, Z.Z, L.L., Y.S., Y.Z., X.Y., Q.B., G.D., and C.L. provided technical help. J.C. provided resources, conceived the project idea and directed the research. L.W., C.L., and J.C. wrote the manuscript.

## Competing interests

The authors declare no competing interests.
