## [Peer Review File · Nature Communications]

REVIEWER COMMENTS

Reviewer #1 (Remarks to the Author):

Perfluoroalkyl Substances (PFAS) are also referred to as Perfluorochemicals (PFCs) are a large family of thousands of synthetic chemicals that are widely used throughout society and found in the environment. They all contain carbon-fluorine bonds, which are one of the most stable chemical bonds and remain in the environment for a long period of time. PFAS has been frequently shown to induce tissue damage and is associated with inflammatory diseases in human and experimental systems. Cellular death and secretion of cytokines including IL-1 β have been previously associated with PFAS exposure. The current study investigates the role of inflammasome and showed that PFOS (one of the common types of PFASs) can engage AIM2 inflammasome and induce IL-1 β production and pyroptosis in macrophages. Mechanistically, they showed that PFOS induced mitochondrial DNA (mtDNA) release and activate AIM2. Moreover, they show that Aim2 deficient (Aim2^{-/-}) mice, but not NLRP3 deficient (Nlrp3^{-/-}) mice, were protected from PFOS-induced tissue inflammation together describing a novel mode of how PFAS can induce inflammatory disorder.

In general, the study is very well structured and executed. The finding that Aim2^{-/-}, but not Nlrp3^{-/-}, mice resist the PFOS-induced inflammation is in my view the central and the most intriguing observation in this study. However, there are some concerns about the physiologic relevance of these findings considering the real environmental exposure to PFAS and the concentrations and experimental setups (IP injection) used in this study. Mechanistic analysis and results aiming to discover how PFOS induces AIM2 inflammasome are at some points incomplete. In particular, the data showed that PFOS involves a number of generic cellular stress responses including JNK, mitochondrial dysfunction and perturbation which then lead to the release of mtDNA and specifically involves AIM2 inflammasomes. Based on all these observations one would suggest that all cellular responses leading to mitochondrial perturbation, BAX/BAK dependent mitochondrial membrane permeabilization or also the therapeutic treatment of cancer patients using venetoclax (Bcl2 antagonist) should also induce AIM2 inflammasome and cause inflammation.

Major points:

1. All experiments are done using mock treatment as control. I was wondering whether any other comparable compound could be used in control groups to show that the observed effects are solely seen, when cells or mice are exposed to PFOS.
2. One central issue is also the concentrations used for treatment of animals and cells. Could authors provide some example in patients with long-term exposure to PFAS. Are the known (detected) concentrations equivalent to the experimental setups in the current study?
3. As already mentioned, authors should show whether JNK activation or mitochondrial damage, induced by other stress cues (including apoptotic and non-apoptotic stimuli), can also induce AIM2 inflammasome.
4. How does PFOS induce the transcriptional upregulation of IL-1 β ?
5. cytosolic mtDNA can also engage cGAC and other inflammatory signalling. Do authors have any data about these signalling machineries?

Minor points

1. Please double-check line 190 referencing the publication 29, which should be 44 (Zhong et al).
2. Line 241-242: "Cytochrome c (cyt C) is a protein normally resided in the inner and outer mitochondrial membrane, participates in the mitochondrial electron-transport chain" cyt C resides in mitochondrial intermembrane space.
3. Line 261-264: "We next found that treatment of THP-1-derived macrophages with MPTP potent inhibitor cyclosporine A (CsA) inhibited the mitochondrial cyt C loss and mtDNA accumulation, indicating that the release of cyt C and mtDNA by PFOS is dependent on MPTP opening (Fig. 4d, e)." this is not supported by Fig. 4D. Please double check the labeling.

Reviewer #2 (Remarks to the Author):

In this ms. Wang and colleagues investigate the effects of Perfluoroalkyl substances (PFAS) on cells, proposing, based on extensive experimentation that PFAS drive an inflammatory response dependent on MPTP, BAX/BAK dependent mtDNA release causing AIM2 dependent inflammasome activity and IL-1 release. They then demonstrate in vivo protection from PFAS inflammatory effects in AIM2 ko mice. The study is timely and offers potential insight into some of the toxic effects of PFAS. In the main, the data support the authors' conclusions, however further clarification of how mtDNA is released from mitochondria is required.

- The authors propose that MPTP is required for mtDNA release, this is based entirely upon CsA expts. They then propose that BAX/BAK are required for mtDNA release. The two processes leading to outer membrane permeabilisation have been shown to be separate events (i.e. MPTP does not require BAX, BAK). Two questions arise, is MPTP actually occurring? (CsA can inhibit multiple targets in addition to CypD, most notably calcineurin), does this step lie upstream of BAX,BAK activity? To test both possibilities, would suggest deletion of cyp D (through CRISPR) should enable this.

- BAX and BAK are usually considered redundant with one another, yet the authors find that deletion of either prevents mtDNA release, which is somewhat unexpected, and should be commented upon. Is deletion of either also preventing cell death in response to PFAS?

Minor point

- Mitotracker deep red (used in 4a) is not a potentiometric dye (unlike TMRE), the reduced signal after treatment suggests loss of mitochondrial mass upon PFAS treatment. This should be commented upon.

Reviewer #3 (Remarks to the Author):

Background: This MS examines the cellular and in vivo mechanisms by which perfluoroalkyl substances (PAFS) trigger inflammation and exacerbation of inflammatory disease in murine models. Due to their surfactant properties, PAFS are present in many consumer products (e.g. stain repellents, paints, non-stick cooking surfaces) but have also been identified as persistent environmental pollutants that can accumulate in living organisms to perturb a wide range of normal biological functions. This study used perfluorooctane sulfonate (PFOS) as a model PAFS which is known to induce accumulation of inflammatory cytokines and cytotoxic responses in multiple tissues, cell types, and animal models.

Experimental Models: Given the ability of PFOS to induce production of IL-1 beta, the authors specifically focus on characterizing the actions of PFOS on inflammasome signaling in: 1) primary murine bone-marrow-derived macrophages (isolated from various inflammasome component genetic knockout strains); 2) human THP1 macrophages (with engineered knockouts of various inflammasome components); 3) an acute (5 day) in vivo model of intraperitoneal PFOS-induced tissue inflammation (in control C57Bl/6 mice and the various inflammasome component knockout strains); and 4) an OVA-induced model of murine asthma with consequent airway/lung inflammation. In the cell-based experiments, they assayed standard readouts of inflammasome signaling including: 1) IL-1b secretion; 2) accumulation of active caspase-1; 3) proIL-1b upregulation and cleavage; 4) ASC oligomerization; 5) gasdermin D cleavage and LDH release (as indices of pyroptotic cell death). In the animal-based experiments, they assayed: 1) inflammatory tissue damage (histology of lungs, liver, kidney); and 2) accumulation of IL-1b, TNF α , and IL-6 in serum, peritoneal fluid, or bronchiolar lavage fluid. In general, the experiments are technically well-designed, are quantified with appropriate statistics, rigor and reproducibility, and are thoughtfully interpreted.

Major findings and conclusions: Based on much supporting data, the authors propose that acute (6 hrs) PFOS treatment induces selective activation of the AIM2 inflammasome (in macrophages) to drive processing/ release of IL-1b and pyroptotic cell death. Because AIM2 is a DNA-sensing inflammasome initiator, additional experiments were performed to support a signaling pathway involving: 1) PFOS-induced upregulation of pro-apoptotic BAX/BAK and downregulation of anti-apoptotic Bcl-2; 2) JNK-mediated activation of BAX/BAK oligomerization and consequent mitochondrial outer membrane permeabilization (MOMP); 3) release of both pro-apoptotic cytochrome C and mitochondrial DNA (mtDNA) into the cytosol; and 4) binding of mtDNA to AIM2 to drive AIM2/ ASC/ caspase-1 inflammasome assembly. Consistent with these ex vivo results, AIM2 knockout mice treated with PFOS (in the acute 5 day peritoneal injection model) exhibit markedly reduced inflammatory damage to their lungs, liver, and kidney that correlates with markedly lower levels of serum/ peritoneal IL-1b but NOT TNFa or IL-6. Likewise, AIM2 knockout mice exposed to PFOS exhibited reduced lung damage and accumulation of inflammatory cytokines in the OVA-induced/PFOS-exacerbated asthma model. Thus, the major finding and conclusion is that activation of the AIM2 inflammasome is a major innate immune response to tissue accumulation of pathogenic amounts of the environmental pollutant PFOS. This is a mechanistically novel finding with regard to the immunological consequences of very high dose perfluoroalkyl substance exposure and tissue accumulation. However, the study lacks broader mechanistic novelty and pathophysiological significance for the specific reasons noted below.

Specific Concerns (Conceptual):

1. Multiple previous studies have demonstrated that activation of the intrinsic (mitochondrial) apoptotic cascade by various sterile stimuli (e.g. acute ionizing radiation-induced genomic DNA damage [PMID2786608; PMID31862870; PMID32760056] or chemotherapeutic drugs [PMID2840962; PMID24078693; PMID31372985] can trigger assembly of AIM2 or NLRP3 inflammasomes in both myeloid and non-myeloid cells. Importantly, a recent study (PMID30965677) by Bae et al demonstrated that circulating cell-free mtDNA in the serum of type 2 diabetic subjects induces AIM2 inflammasome activation in macrophages (after internalization) to drive chronic sterile inflammation. Thus, the finding that primary induction of intrinsic apoptosis by PFOS can induce AIM2 inflammasome assembly and sterile inflammation is an incremental advance.
2. It's clear that primary responses to high-dose PFOS exposure are: 1) activation of NFkB signaling (by an undefined mechanism) which induces not only proIL-1b expression as inflammasome-dependent inflammatory cytokine, but also TNFa and IL-6 as inflammasome-independent cytokines; 2) transcriptional regulation of pro-and anti-apoptotic Bcl2-family members by undefined signaling pathways; and 3) the acute activation of JNK signaling (by an undefined mechanism) to induce oligomerization of the upregulated BAX and BAK. Induction of AIM2 inflammasome assembly (in macrophages) is a secondary (albeit significant) consequence of exposure to high dose PFOS. As noted, the mechanisms by which high-dose PFOS triggers these primary responses (upstream of AIM2 activation) are not investigated in this study. It can be reasonably argued that characterization of the NFkB activation mechanism is beyond the scope of the present study. However, delineation of the underlying mechanism (s) for the primary intrinsic apoptotic induction is central to understanding how two regulated cell death pathways (primary AIM2-independent apoptosis and secondary AIM2-dependent pyroptosis), which are triggered by PFOS, are temporally and spatially integrated at the cellular and systemic levels to coordinate systemic inflammation in mice (or humans) exposed to low-dose or high-dose PFOS. Minimally, experiments that explore the presumed post-transcriptional mechanism(s) by which PFOS triggers JNK activation and JNK-dependent BAX/BAK oligomerization are required.
3. As noted in the paper's introduction and discussion, the broad exposure of most human subjects to environmental PFOS is limited to consumer products and results in serum levels of ~100 ng/ml. Given PFOS's molecular mass of 500, this translates to 0.2 uM concentration which is a 1000-fold lower than the 100-300 uM concentrations used in the cell culture experiments of this study. The authors acknowledge that the latter concentrations would translate to serum concentrations of ~90,000 ng/ml which may only be observed in some human subjects (such as fluorochemical production workers) exposed to extraordinarily high levels of environmental PFOS. This raises important issues regarding the broader pathophysiological/ clinical significance of the observed

AIM2 inflammasome pathway in understanding the mechanisms by which chronic inflammatory disease and tissue dysfunction is induced by the much lower PFOS environmental exposure to consumer products in general human population cohorts.

Specific Concerns (Experimental):

1. The cell death analyses in control and AIM2-ko macrophages are limited to 6 hr PFOS treatments. Longer term treatments with PFOS should be included (particularly in the AIM2-ko cells) to analyze: 1) the kinetics of progression to secondary necrosis/lytic cell death; 2) roles of secondary GSDMD-independent but GSDME-dependent pyroptosis. (GSDME can be cleaved and activated by the apoptotic executioner caspase-3.)

2. Related to above concern, the cell death and AIM2 experiments are limited to macrophages. However, AIM2 inflammasome signaling and cell death in non-myeloid cells, particularly epithelial cells (PMID27846608; PMID29973713; PMID31919014), can be a major driver of acute or chronic inflammatory responses to sterile stimuli. Experiments that analyze PFOS-induced AIM2 inflammasome signaling in an epithelial cell model should be included to establish whether PFOS drives similar AIM2 responses in non-myeloid cells.

3. Related to the above issue of non-myeloid AIM2 inflammasome signaling: The *in vivo* experiments described in Figures 6 and 7 shows that: 1) PFOS-induced tissue inflammation is suppressed in AIM2-deficient mice; and 2) PFOS-induced accumulation of IL-1b, but not TNF α or IL-6, is also suppressed in the AIM2-deficient mice. This raises very relevant questions as to whether production of IL-1b is, or is not, the major driver of inflammatory tissue damage in the PFOS-treated wildtype mice. Experiments using IL-1b-knockout or IL-1 receptor-knockout mice are required to address this important mechanistic issue. It is possible that AIM2-dependent pyroptosis and/or AIM2-dependent IL-18 production, independently of AIM2-dependent IL-1b accumulation, may be the major driver of inflammation in the PFOS-exposed mice.

4. Figure 6b,c,d and Extended Data Figure 1b,c,d and Methods, lines 581-583: The 3 right-most subpanels in each of these figure panels and the relevant Methods section indicates: "At 5-days post-treatment mice liver, lung and kidney tissue were isolated and cultured for 24 hr, and the supernatants were analyzed by ELISA." More detail is required to explain how the liver, lung and kidney was processed for these *ex vivo* incubations and analyses. For example, were tissue slices prepared? If so, how thin were the slices? If not sliced, were crude cell suspensions prepared? If the latter, how were these cell suspensions generated? With tissue digesting enzymes?

Dear Referees,

Thank you very much for providing us a valuable opportunity to revise our paper entitled “*Innate immune activation through AIM2 inflammasome sensing of perfluoroalkyl substances*”. We really appreciate the positive and thoughtful comments regarding our manuscript, and we have performed additional experiments and now present new data addressing all of the critical points raised by the Referees. A point-by-point response to the Referees’ concerns is included below.

We hope that the revised manuscript meets the requirements for publication in “*Nature Communications*”, and we look forward to hearing from you.

Sincerely,

Jun Cui, Ph.D.

Professor and Dean of Department of Biochemistry

School of Life Sciences

Sun Yat-sen University

(The corresponding author)

Response to the comments of Reviewer#1

In general, the study is very well structured and executed. The finding that Aim2^{-/-}, but not Nlrp3^{-/-}, mice resist the PFOS-induced inflammation is in my view the central and the most intriguing observation in this study. However, there are some concerns about the physiologic relevance of these findings considering the real environmental exposure to PFAS and the concentrations and experimental setups (IP injection) used in this study. Mechanistic analysis and results aiming to discover how PFOS induces AIM2 inflammasome are at some points incomplete. In particular, the data showed that PFOS involves a number of generic cellular stress responses including JNK, mitochondrial dysfunction and perturbation which then lead to the release of mtDNA and specifically involves AIM2 inflammasomes. Based on all these observations one would suggest that all cellular responses leading to mitochondrial perturbation, BAX/BAK dependent mitochondrial membrane permeabilization or also the therapeutic treatment of cancer patients using venetoclax (Bcl2 antagonist) should also induce AIM2 inflammasome and cause inflammation.

Major points:

Comment 1: *All experiments are done using mock treatment as control. I was wondering whether any other comparable compound could be used in control groups to show that the observed effects are solely seen, when cells or mice are exposed to PFOS.*

Response: As the matter of fact, at the beginning of this project, we also detected the IL-1 β production in PFOA (another comparable compound)-treated THP-1-derived macrophage. However, the results showed that PFOA could not promote the production of IL-1 β in THP-1-derived macrophages (see **New Figure 1a-b**), suggesting that different kinds of PFAS may have different effects on cells basing on its distinct chemical structure and composition.

New Figure 1: **a** THP-1-derived macrophages were stimulated with PFOA as indicated for 6 h. Cell supernatants were collected to determine IL-1 β production. **b** THP-1-derived macrophages were stimulated with PFOA (50 μ M) or PFOS (50 μ M) for 6 h. Cell supernatants were harvested to determine IL-1 β production. In **a** and **b**, data are mean values \pm SEM, ns (non-significant), $P > 0.05$; *** $P < 0.001$ (Student's t test).

Comment 2: *One central issue is also the concentrations used for treatment of animals and cells. Could authors provide some example in patients with long-term exposure to PFAS. Are the known (detected) concentrations equivalent to the experimental setups in the current study?*

Response: Numerous population-based epidemiological studies have indicated that PFAS is a significant risk factor for a variety of human diseases¹⁻⁵. There are also some patient-based studies, which have reported the association of PFAS exposure with clinical/pathological parameters of certain diseases. For example, the increase of serum PFAS levels are found in: 1) liver fibrosis in nonalcoholic fatty liver disease in children (PFOS level, 0.53-4.35 ng/ml)⁶; 2) celiac disease in children (PFOS level, median 2.02 ng/ml)⁷; 3) hepatocellular carcinoma (PFOS level, 4.36-48.4 ng/ml)⁸; and 4) cirrhosis (PFOS level, 1.12-126 ng/ml)⁸. These clinical studies imply that the

concentrations of serum PFAS seems variable in patients with different diseases. In addition, some epidemiological studies have shown that occupational exposure workers had higher serum PFAS levels, for example, Olsen *et al.* reported that serum PFOA concentration in fluorochemical production workers ranged from 7 to 92030 ng/ml (median 1100 ng/ml)⁹; Zhou *et al.* reported that serum PFOS concentration in fishery employees ranged from 82.6 to 31400 ng/ml (median 10400 ng/ml)¹⁰. However, these highly exposed workers didn't appear significant clinical symptoms, and such phenomenon indicates these workers may be tolerated to PFAS exposure. Taken together, although the epidemiological and clinical studies have confirmed the adverse effects of PFAS on human health, so far there is no exact "threshold" or "range" for PFAS exposure levels related to certain disease syndrome. Indeed, we have measured the serum PFOS level in the same mouse model (C57BL/6) in another study and the results showed that PFOS concentration in mice exposed to 7 mg/kg for 28 days (total 196 mg/kg) was about 8430 ng/ml¹¹. Using this PFOS value, we can calculate that every 1 mg/kg PFOS exposure in mouse is equivalent to 43 ng/ml (8430/196) serum PFOS concentration. Hence, in the current study, we can speculate that the serum PFOS levels in acute exposure model (5, 15, and 25 mg/kg for 5 days) are 1075, 3225, and 5375 ng/ml, respectively; while PFOS level in chronic exposure model (0.066 mg/kg for 30 days; 5 mg/kg for 20 days) is about 86 ng/ml and 4300 ng/ml, respectively. We found that wild type (WT) mice exposure to PFOS (0.066 mg/kg per day for 30 days) could also induce chronic inflammatory responses and tissue damage (see **New Figure 2a, related to Extended Data Figure 2a in the**

Supplemental material). The proinflammatory cytokines IL-1 β , TNF- α and IL-6 were significantly increased in the serum, peritoneal fluids, liver, lung and kidney in the PFOS-treated mice, compared to those in control mice (see **New Figure 2b-d, related to Extended Data Figure 2b-d in the Supplemental material**). These results demonstrate that lower concentration of PFOS could also induce inflammatory cytokines production and tissue injury *in vivo*, which are in line with the acute exposure or asthma exacerbation model (see **Extended Data Figure 1 in the Supplemental material and Figure 7 in the manuscript**). Therefore, we consider the PFOS administration dosage in current animal experiments could be comparable to the occupational exposure level in human; and our experimental dosage is lower than many other PFOS animal studies (usually using more than 200 mg/kg totally in both acute and chronic exposure models)¹²⁻¹⁴. The serum level of PFOS in most human subjects ranges from 0.2 to 99.7 ng/ml¹⁵⁻¹⁶, and this translates to about 200 nM concentration (PFOS's molecular mass is 538.22), so we treated THP-1-derived macrophages with suggested dose of PFOS (200 nM) in time-dependent manner, and found that this concentration of PFOS could also induce pro-inflammatory cytokines release (IL-1 β , TNF- α , and IL-6) and cell death (see **New Figure 3a, related to Extended Data Figure 6a in the Supplemental material**). In addition, the much lower concentration range (10-200 nM) of PFOS could also trigger inflammatory cytokines production (IL-1 β , TNF- α and IL-6) and cell death after PFOS exposure 48 h post stimulation (see **New Figure 3b, related to Extended Data Figure 6b in the Supplemental material**). Moreover, *AIM2* deficient THP-1 macrophages could

significantly decrease IL-1 β production and cell death induced by PFOS (200 nM, 48 h) (see **New Figure 3c, related to Extended Data Figure 8e-f in the Supplemental material**). Taken together, the above *in vivo* chronic exposure model and *in vitro* cell experiments with low concentration of PFOS, which represents the exposure level in general human populations, can also induce chronic inflammatory responses and tissue damage in animal and cell culture system in an AIM2-dependent manner. We have added the above information in the Discussion part.

New Figure 2: a-d C57BL/6 mice (6 weeks old) were treated with PFOS (0.06 mg/kg body weight per day dissolved in PBS containing 2 % Tween-80) (n=5) or treated with PBS containing 2 % Tween-80 (Mock, n=5) by intraperitoneal injection for 5 days. At 5 days post-treatment, mice liver, lung and kidney tissue were stained with hematoxylin-eosin (H&E) and assayed using a light microscope (**a**). Scale bar, 100 μ m. The tissue (liver, lung, and kidney) injury score was determined in 5 randomly selected nonoverlapping fields from respective individual mouse tissue sections at x400 magnification. The scores were averaged in respective organs (liver, lung, and kidney) and individual animals. All histology analyses were conducted in a

blinded manner. At 5 days post-treatment, the liver, lung and kidney of treated mice were isolated and cultured for 24 h, and supernatants were analyzed by ELISA for IL-1 β (b), TNF- α (c) and IL-6 (d). In (a-d), data are mean values \pm SD, ** P < 0.01; ***P < 0.001 (Student's t test).

New Figure 3: a-b THP-1-derived macrophages were treated with PFOS (200 nM) in a time-dependent manner (a) or treated with PFOS in a dose-dependent manner for 48

h (b). The production of IL-1 β , TNF- α and IL-6 and cell death were determined. c *AIM2* KO THP-1-derived macrophages were treated with PFOS (200 nM, 48 h). IL-1 β and LDH release was measured in the supernatants. In a-c, data are mean values \pm SEM, ns (non-significant), $P > 0.05$; * $P < 0.05$; *** $P < 0.001$ (Student's t test).

Comment 3: As already mentioned, authors should show whether JNK activation or mitochondrial damage, induced by other stress cues (including apoptotic and non-apoptotic stimuli), can also induce *AIM2* inflammasome.

Response: As the reviewer suggested, we determined the function of *AIM2* inflammasome under other stimuli (cholesterol-water soluble, which could induce mitochondrial damage; and surfactin, which could trigger JNK activation) by knocking out *AIM2* in the THP-1-derived macrophages. The results showed that cholesterol-water soluble or surfactin-induced IL-1 β production was significantly decreased in *AIM2* KO cells (see **New Figure 4**). And the similar results have been reported to show that cholesterol-water soluble induces mtDNA release to trigger *AIM2* inflammasome activation and surfactin-induced JNK activation is responsible for *AIM2* inflammasome activation^{17,18}.

New Figure 4: Wild type (WT) or *AIM2* knockout (KO) THP-1-derived macrophages were pre-treated in culture medium with LPS (200 ng/ml) for 3 h, followed by ATP (5 mM, 6 h), poly(dA:dT) (2 µg/ml, 6 h), cholesterol- water soluble (100 µg/ml, 6 h) and surfactin (5µg/ml, 6 h). Cell supernatants were collected to measure IL-1β release by ELISA. Data are mean values ± SEM, ns (non-significant), P > 0.05; ** P < 0.01; *** P < 0.001 (Student's t test).

Comment 4: *How does PFOS induce the transcriptional upregulation of IL-1β?*

Response: As shown in the manuscript, PFOS significantly increased the secretion of pro-inflammatory cytokines (IL-1β, TNF-α and IL-6), indicating that PFOS could induce the activation of NF-κB signaling pathway. Next, we investigated the mechanism underlying PFOS-induced NF-κB signaling activation. In BMDMs, PFOS exposure triggered the phosphorylation of NF-κB p65 subunit and the degradation of NF-κB inhibitor alpha (IκBα) in a dose- or time-dependent manner (see **New Figure 5a, related to Extended Data Figure 4c in the Supplemental material; New Figure 5b**). The increased Ca²⁺ in cytosolic compartment ($[Ca^{2+}]_c$) is essential for downstream calcium-dependent signaling pathway activation (such as protein kinase C, followed by that of downstream of and NF-κB signaling activation) and Ca²⁺ signaling can be triggered by the release of Ca²⁺ from intracellular endoplasmic reticulum (ER) storage. Hence, we analyzed the effect of PFOS on $[Ca^{2+}]_c$ and found that $[Ca^{2+}]_c$ and the protein level of ER stress monitor Bip were significantly increased in PFOS-treated BMDMs (see **New Figure 5c, related to Extended Data Figure 4d in the Supplemental material; New Figure 5d**). To further determine the

involvement of $[Ca^{2+}]_c$ and PKC in PFOS-triggered NF- κ B signaling activation, we pretreated BMDMs with Ca^{2+} chelator (BAPTA-AM) or PKC inhibitor (Chelerythrine Chloride) before PFOS treatment, and we found that phosphorylation of p65, the degradation of I κ B α , and the pro-inflammatory cytokines (IL-1 β , TNF- α and IL-6) were evidently inhibited in the presence of BAPTA-AM or PKC inhibitor (see **New Figure 5e-g, related to Extended Data Figure 4e-g in the Supplemental material**).

Consistent with these observations, the similar results were observed in THP-1-derived macrophages (see **New Figure 5h, 5j and 5l-n, related to Extended Data Figure 5c-g in the Supplemental material; New Figure 5i and 5k**).

Collectively, Ca^{2+} -PKC dependent pathway was critical for PFOS-mediated NF- κ B signaling activation, which leads to the transcription and production of pro-IL-1 β , TNF- α and IL-6.

New Figure 5: Immunoblot analysis of NF- κ B signaling pathway in BMDMs treated as indicated. BMDMs (**a-b**) or THP-1-derived macrophages (**h-i**) were treated with PFOS in a dose- (**a,h**) or time- (**b,i**) dependent manner. The cytosolic Ca²⁺ ([Ca²⁺]_c) in BMDMs (**c-d**) or THP-1-derived macrophages (**j-k**) treated with PFOS as indicated does and time, assessed by fluorescence of cells stained with Fluo-4 AM via using fluorescence microplate reader (excitation and emission wavelength at 488 nm and 520 nm, respectively). The cytosolic Ca²⁺ ([Ca²⁺]_c) was presented as the percentage relative to the value in control culture. BMDMs (**e-g**) or THP-1-derived macrophages (**l-n**) were pretreated with Ca²⁺ chelator (BAPTA-AM, 50 μ M for 1 h) or PKC inhibitor Chelerythrine Chloride (Ch-Chloride, 10 μ M for 1 h) and subsequently treated with PFOS (150 μ M, 6 h). Cell lysates were collected for immunoblot analysis. Cell supernatants were harvested to determine the release of proinflammatory cytokines (IL-1 β , TNF- α and IL-6) by ELISA. Data are mean values \pm SEM, ns (non-significant), P > 0.05; ** P < 0.01; *** P < 0.001 (Student's t test).

Comment 5: *cytosolic mtDNA can also engage cGAS and other inflammatory signalling. Do authors have any data about these signalling machineries?*

Response: As the reviewer suggested, we checked the type I interferon (IFN-I) pathway and inflammasome pathway via detecting IFN- β production and IL-1 β when the THP-1-derived macrophages were treated with PFOS, since cGAS could sense mtDNA to induce IFN- β production. However, we can only detect the release of IL-1 β , but not IFN- β in the cells treated with PFOS in dose- or time-dependent manner (see **New Figure 6a** and **Figure 1c in the manuscript**). Hence, we investigated that whether PFOS had ability to inhibit the activation of IFN-I pathway. Strikingly, we found that PFOS could inhibit the Sendai virus (SeV)-induced IFN-I signaling (see **New Figure 6b**). Therefore, PFOS, on one hand, triggered AIM2 inflammasome-dependent IL-1 β production. On the other hand, PFOS suppressed IFN-I pathway activation through undiscovered mechanisms, which would be of interest to determine the detailed mechanism in future.

New Figures 6: **a** THP-1-derived macrophages were treated with PFOS as indicated, cell supernatants were collected to detect IFN- β production by ELISA. **b** HEK293T cells were transfected with an IFN- β -luc reporter plasmid. After 12 h, cells were treated with SeV (MOI = 0.1) for 24 h, and subsequently treated with PFOS in dose-dependent manner for 6 h. Cell lysates were collected to determine the IFN- β -luc activity. Data are mean values \pm SEM, ns (non-significant), $P > 0.05$ (Student's t test), *** $P < 0.001$ (Student's t test).

Minor points:

Comment 6: Please double-check lane 190 referencing the publication 29, which should be 44 (Zhong et al).

Response: We thank the reviewer for noticing this and changed the reference in the manuscript.

Comment 7: Lane 241-242: "Cytochrome c (cyt C) is a protein normally resided in the inner and outer mitochondrial membrane, participates in the mitochondrial electron-transport chain" cyt C resides in mitochondrial intermembrane space.

Response: We thank the reviewer for noticing this and changed the description of cytochrome c (cyt C) as “cyt C is a protein normally resided in the mitochondrial intermembrane space, participates in the mitochondrial electron-transport chain” in the manuscript.

Comment 8: Lane 261-264: “We next found that treatment of THP-1-derived macrophages with MPTP potent inhibitor cyclosporine A (CsA) inhibited the mitochondrial cyt C loss and mtDNA accumulation, indicating that the release of cyt C and mtDNA by PFOS is dependent on MPTP opening (Fig. 4d, e).” this is not supported by Fig. 4D. Please double check the labeling.

Response: We have changed it accordingly.

Response to the comments of Reviewer#2

In this ms. Wang and colleagues investigate the effects of Perfluoroalkyl substances (PFAS) on cells, proposing, based on extensive experimentation that PFAS drive an inflammatory response dependent on MPTP, BAX/BAK dependent mtDNA release causing AIM2 dependent inflammasome activity and IL-1 release. They then demonstrate in vivo protection from PFAS inflammatory effects in AIM2 ko mice. The study is timely and offers potential insight into some of the toxic effects of PFAS. In the main, the data support the authors' conclusions, however further clarification of how mtDNA is released from mitochondria is required.

Major points:

Comment 1: The authors propose that MPTP is required for mtDNA release, this is based entirely upon CsA expts. They then propose that BAX/BAK are required for

mtDNA release. The two processes leading to outer membrane permeabilisation have been shown to be separate events (i.e. MPTP does not require BAX, BAK). Two questions arise, is MPTP actually occurring? (CsA can inhibit multiple targets in addition to CypD, most notably calcineurin), does this step lie upstream of BAX,BAK activity ? To test both possibilities, would suggest deletion of cyp D (through CRISPR) should enable this.

Response: In our study, we found that the treatment of indicated cells with MPTP potent inhibitor cyclosporine A (CsA) apparently inhibited PFOS-induced caspase-1 activation and IL-1 β maturation/secretion (see **Figure 4d-f in the manuscript**). To confirm that MPT actually happened, we knocked out cyclophilin D (CypD), a critical component of the MPTP and found that *CypD* KO THP-1-derived macrophages showed partly but significantly decreased mtDNA release, IL-1 β secretion, and maturation in response to PFOS treatment, indicating that MPTP is actually involved in PFOS-induced mtDNA release (see **New Figure 7a, related to Figure 4g-h in the manuscript**). Furthermore, we observed that CsA impaired the PFOS-induced release of cyt C and mtDNA in the *CypD*-KO-derived macrophages (see **New Figure 7b, related to Figure 4i-j in the manuscript**), suggesting that in addition to its effect on MPTP, CsA might also suppress PFOS-induced cyt C and mtDNA release via additional mechanism (most likely MOMP). Taken together, these results indicate that PFOS exposure induces MPTP opening as well as non-MPTP pathway, allowing cyt C and mtDNA release into the cytosolic compartment. It was previously reported that BAX which translocates to mitochondria, can interact with CypD to promote MPTP formation¹⁹⁻²². We found that PFOS exposure could trigger the interaction between

BAX and CypD (see **New Figure 7c, related to Extended Data Figure 12d in the Supplemental material**). Additionally, the mtDNA release, cell death, and IL-1 β production/maturation were significantly decreased by knockout of either BAX or BAK, suggesting that both BAX and BAK contributed to the MOMP (see **New Figure 7d, related to Figure 5e-h in the manuscript**). Moreover, the single BAX and BAK deficiency showed more mtDNA release, cell death and IL-1 β secretion/maturation than BAX/BAK double KO, suggesting that BAX/BAK play important roles in PFOS-induced inflammasome activation (see **New Figure 7d, related to Figure 5e-h in the manuscript**). Collectively, our data illustrated that PFOS triggers the mtDNA release through BAX/BAK-mediated MPTP and BAX/CypD-mediated MOMP pathway, resulting in AIM2 inflammasome activation.

New Figure 7: a Wild type (WT), *CypD* knockout (KO) THP-1-derived macrophages were treated with PFOS (150 μ M) for 6 h. Relative enrichment of mtDNA in AIM2-pulldown material were determined. IL-1 β production in the supernatants were measured by ELISA. The maturation of IL-1 β in the supernatants or pro-IL-1 β and *CypD* in lysates were detected by immunoblot. **b** Immunoblot analysis of cytochrome c from cytosolic extracts and mitochondria derived from the cells pre-treated with DMSO or cyclosporine A (50 μ M) for 1 h. WT and *CypD* KO THP-1-derived macrophages were then treated with PFOS (150 μ M, 6 h) as indicated. Relative enrichment of mtDNA in AIM2-pulldown material from the THP-1-derived macrophages as indicated. **c** THP-1-derived macrophages were treated with PFOS (150 μ M, 6 h) and then cell lysates were subjected to immunoprecipitation with anti-*CypD* and immunoblot with anti-BAX. **d** WT, *BAX* KO, *BAK* KO, *BAX/BAK* KO-THP-1-derived macrophages were treated with PFOS (150 μ M, 6 h). Relative enrichment of mtDNA in AIM2-pulldown material were determined. Cell death was determined by detecting LDH release. IL-1 β production in the supernatants were measured by ELISA. The maturation of IL-1 β in the supernatants or pro-IL-1 β in lysates were detected by immunoblot. In **a**, **b** and **d**, data are mean values \pm SEM, ns (non-significant), $P > 0.05$; ** $P < 0.01$; *** $P < 0.001$ (Student's t test).

Comment 2: *BAX* and *BAK* are usually considered redundant with one another, yet the authors find that deletion of either prevents mtDNA release, which is somewhat unexpected, and should be commented upon. Is deletion of either also preventing cell death in response to PFAS?

Response: Several evidences have supported that BAX acts as a pivotal role in regulating MPT and MOMP via interacting with CypD and BAK, respectively¹⁹⁻²⁵. We have found that PFOS exposure could trigger the interaction between BAX and CypD (see **New Figure 7c, related to Extended Data Figure 12d in the Supplemental material**) and *CypD* KO THP-1-derived macrophages showed partly but significantly decreased mtDNA release, IL-1 β secretion, and maturation in response to PFOS treatment (see **New Figure 7a, related to Figure 4g-h in the manuscript**). Taken together, these results indicated that BAX interacted with CypD to mediate MPTP opening in response to PFOS. Additionally, we found that BAX or BAK single deletion could partly but significantly reduced the mtDNA release and cell death (see **New Figure 7d, related to Figure 5e-h in the manuscript**). And the similar phenomenon was observed in previous reports²⁶⁻²⁷. Moreover, BAX and BAK double deletion almost eliminated the mtDNA release and showed more less cell death (see **New Figure 7d, related to Figure 5e-h in the manuscript**). Collectively, these results suggested that the mtDNA released into cytosolic compartment through two non-mutually exclusive pathways: BAX interacts with CypD to form MPTP and BAX/BAK oligomerize to form MOMP, thereby resulting in cell death and AIM2 inflammasome activation. Hence, given that PFOS-induced mtDNA release through two BAX-dependent pathways, it would be to understand why deletion BAX or BAK could decrease the mtDNA release.

Minor point:

Comment 3: *Mitotracker deep red (used in 4a) is not a potentiometric dye (unlike TMRE), the reduced signal after treatment suggests loss of mitochondrial mass upon PFAS treatment. This should be commented upon.*

Response: We thank the reviewer for noticing this and confirmed the description accordingly. As a matter of fact, the reduced signal of Mitotracker deep red and Mitotracker green suggested that the decreased mitochondrial respiration and mitochondrial mass, respectively^{17,28}.

Response to the comments of Reviewer#3

Major findings and conclusions: Based on much supporting data, the authors propose that acute (6 hrs) PFOS treatment induces selective activation of the AIM2 inflammasome (in macrophages) to drive processing/ release of IL-1 β and pyroptotic cell death. Because AIM2 is a DNA-sensing inflammasome initiator, additional experiments were performed to support a signaling pathway involving: 1) PFOS-induced upregulation of pro-apoptotic BAX/BAK and downregulation of anti-apoptotic Bcl-2; 2) JNK-mediated activation of BAX/BAK oligomerization and consequent mitochondrial outer membrane permeabilization (MOMP); 3) release of both pro-apoptotic cytochrome C and mitochondrial DNA (mtDNA) into the cytosol; and 4) binding of mtDNA to AIM2 to drive AIM2/ ASC/ caspase-1 inflammasome assembly. Consistent with these ex vivo results, AIM2 knockout mice treated with PFOS (in the acute 5 days peritoneal injection model) exhibit markedly reduced inflammatory damage to their lungs, liver, and kidney that correlates with markedly lower levels of serum/ peritoneal IL-1 β but NOT TNF- α or IL-6. Likewise, AIM2 knockout mice exposed to PFOS exhibited reduced lung damage and accumulation of inflammatory cytokines in the OVA-induced/PFOS-exacerbated asthma model. Thus, the major finding and conclusion is that activation of the AIM2 inflammasome is a

major innate immune response to tissue accumulation of pathogenic amounts of the environmental pollutant PFOS. This is a mechanistically novel finding with regard to the immunological consequences of very high dose perfluoroalkyl substance exposure and tissue accumulation. However, the study lacks broader mechanistic novelty and pathophysiological significance for the specific reasons noted below.

Specific Concerns (Conceptual):

Comment 1: *Multiple previous studies have demonstrated that activation of the intrinsic (mitochondrial) apoptotic cascade by various sterile stimuli (e.g. acute ionizing radiation-induced genomic DNA damage [PMID:2786608; PMID31862870; PMID32760056] or chemotherapeutic drugs [PMID:2840962; PMID:24078693; PMID31372985] can trigger assembly of AIM2 or NLRP3 inflammasomes in both myeloid and non-myeloid cells. Importantly, a recent study (PMID30965677) by Bae et al demonstrated that circulating cell-free mtDNA in the serum of type 2 diabetic subjects induces AIM2 inflammasome activation in macrophages (after internalization) to drive chronic sterile inflammation. Thus, the finding that primary induction of intrinsic apoptosis by PFOS can induce AIM2 inflammasome assembly and sterile inflammation is an incremental advance.*

Response: Thanks for this suggestion. We have discussed several studies reporting the roles of NLRP3 and AIM2 involved in sterile stimuli (mtDNA and genomic DNA)-mediated inflammation (Discussion part, 3rd paragraph)²⁹⁻³¹. We have also added the discussion about the findings reported by Bae et al³². It seems that there are not many studies reporting mtDNA sensed by AIM2 inflammasome; in addition, the detailed mechanism of mtDNA release was not elucidated in previous works. Recently, only a few studies have investigated the association of inflammasome and environmental stimulants (such as silica, particulate matters, and nanoparticles), and they have mainly focused on NLRP3 inflammasome³³⁻³⁵. Until now, the host sensing mechanisms in response to organic toxins have not been reported yet. Our study for the first time demonstrated that AIM2 activation is required for PFAS-induced acute or chronic inflammatory response and tissue damage, in which the process is

dependent on mitochondrial dysfunction and release of mtDNA. Based on the suggestions of the reviewer, we performed a series of new experiments and revealed that PFOS triggered the mtDNA release through BAX/CypD-mediated MOMP and BAX/CypD-mediated MPTP pathway, resulting in AIM2 inflammasome activation but not NLRP3 inflammasome. Nonetheless, our findings reveal that mtDNA-AIM2 axis is a critical innate mechanism for PFAS-induced inflammation (see **New Figure 8, related to Figure 8 in the manuscript**), and may have important implications for the development of therapy against PFAS-induced toxicity and tissue damage.

New Figure 8: Schematic diagram to illustrate the mechanisms that PFOS exposure induces AIM2 inflammasome activation to promote tissue damage. PFOS induces mitochondrial DNA (mtDNA) release through BAX-CypD-mediated mitochondrial permeability-transition pores and BAX/BAK mediated macropores; and PFOS-induced mtDNA serves as an essential danger-associated molecular pattern (DAMP) to trigger AIM2 inflammation activation, ultimately leading to tissue inflammation.

Comment 2: *It's clear that primary responses to high-dose PFOS exposure are: 1) activation of NFκB signaling (by an undefined mechanism) which induces not only proIL-1β expression as inflammasome-dependent inflammatory cytokine, but also TNFα and IL-6 as inflammasome-independent cytokines; 2) transcriptional regulation of pro-and anti-apoptotic Bcl2-family members by undefined signaling pathways; and 3) the acute activation of JNK signaling (by an undefined mechanism) to induce oligomerization of the upregulated BAX and BAK. Induction of AIM2 inflammasome assembly (in macrophages) is a secondary (albeit significant) consequence of exposure to high dose PFOS. As noted, the mechanisms by which high-dose PFOS triggers these primary responses (upstream of AIM2 activation) are not investigated in this study. It can be reasonably argued that characterization of the NFκB activation mechanism is beyond the scope of the present study. However, delineation of the underlying mechanism (s) for the primary intrinsic apoptotic induction is central to understanding how two regulated cell death pathways (primary AIM2-independent apoptosis and secondary AIM2-dependent pyroptosis), which are triggered by PFOS, are temporally and spatially integrated at the cellular and systemic levels to coordinate systemic inflammation in mice (or humans) exposed to low-dose or high-dose PFOS. Minimally, experiments that explore the presumed post-transcriptional mechanism(s) by which PFOS triggers JNK activation and JNK-dependent BAX/BAK oligomerization are required.*

Response: In BMDMs and THP-1-derived macrophages, PFOS exposure triggered the phosphorylation of NF-κB p65 subunit and the degradation of NF-κB inhibitor alpha (IκBα) in a dose- or time-dependent manner (see **New Figure 5a, related to Extended Data Figure 4c in the Supplemental material; New Figure 5b; New Figure 5h, related to Extended Data Figure 5c in the Supplemental material; New Figure 5i**). The increased Ca²⁺ in cytosolic compartment ([Ca²⁺]_c) is essential for downstream calcium-dependent signaling pathway activation (such as protein

kinase C, followed by downstream of NF- κ B signaling and JNK activation) and Ca^{2+} signaling can be triggered by the release of Ca^{2+} from intracellular endoplasmic reticulum (ER) storage. Hence, we analyzed the effect of PFOS on $[\text{Ca}^{2+}]_c$ and found that $[\text{Ca}^{2+}]_c$ and the protein level of ER stress monitor Bip were significantly increased in PFOS-treated BMDMs and THP-1-derived macrophages (see **New Figure 5c, related to Extended Data Figure 4d in the Supplemental material; New Figure 5d; New Figure 5j, related to Extended Data Figure 5d in the Supplemental material; New Figure 5k**). To further determine the involvement of $[\text{Ca}^{2+}]_c$ and PKC in PFOS-triggered NF- κ B signaling activation, we performed experiments with indicated inhibitors, and found that phosphorylation of p65, the degradation of I κ B α , the pro-inflammatory cytokines (IL-1 β , TNF- α and IL-6) and phosphorylation of JNK were evidently inhibited in the presence of BAPTA-AM or PKC inhibitor (see **New Figure 5e-g, related to Extended Data Figure 4e-g in the Supplemental material; 5l-n, related to Extended Data Figure 5c-g in the Supplemental material; New Figure 9a; New Figure 9b, related to Extended Data Figure 12c in the Supplemental material**). Collectively, our data demonstrated that Ca^{2+} -PKC dependent pathway was critical for PFOS-mediated NF- κ B signaling activation and JNK activation.

We agree with the reviewer's comments that there might be a kinetics dependency between PFOS-induced AIM2-independent apoptosis and AIM2-dependent pyroptosis. In this study, we detected the PFOS-induced apoptosis with Annexin V/7AAD staining by flow cytometry analysis and the PFOS-induced

pyroptosis using immunoblot analysis, respectively. We found that PFOS could trigger evident apoptosis and pyroptosis in a dose-dependent manner (see **Figure 1d in the manuscript and Extended Data Figure 11e in the Supplemental material**). Moreover, we found that in macrophages depleted mtDNA (ρ^0) with PFOS treatment still underwent apoptosis (see **New Figure 9c**). However, the PFOS-induced pyroptosis was nearly eliminated in ρ^0 macrophages (see **New Figure 9d**). In addition, we knocked down *caspase-3* in the THP-1-derived macrophages, and found that the cleavage of GSDMD was not impaired in *caspase3* KD THP-1-derived macrophages, suggesting that PFOS-induced caspase3-mediated apoptosis is irresponsible for AIM2-dependent pyroptosis (see **New Figure 9e**). Collectively, caspase-3-mediated apoptosis and AIM2 inflammasome-dependent pyroptosis are two separate cell death pathways.

New Figure 9: BMDMs (a) or THP-1-derived macrophages (b) were pretreated with Ca^{2+} chelator (BAPTA-AM, 50 μM for 1 h) or PKC inhibitor (Chelerythrine Chloride, 10 μM for 1 h) and subsequently treated with PFOS (150 μM , 6 h). Cell lysates were collected for Immunoblot analysis. Apoptosis was assayed with Annexin V/7AAD staining by flow cytometry analysis (c) and immunoblot analysis of PFOS-induced pyroptosis of deleted mtDNA THP-1 cells (ρ^0) with PFOS (150 μM , 6 h) (d). e *caspase-3* knockdown (KD) THP-1-derived macrophages were treated with PFOS (150 μM , 6 h). Cell lysates were collected to determine the cleavage of GSDMD for analyzing PFOS-induced pyroptosis.

Comment 3: As noted in the paper's introduction and discussion, the broad exposure of most human subjects to environmental PFOS is limited to consumer products and results in serum levels of ~ 100 ng/ml. Given PFOS's molecular mass of 500, this

translates to 0.2 uM concentration which is a 1000-fold lower than the 100-300 uM concentrations used in the cell culture experiments of this study. The authors acknowledge that the latter concentrations would translate to serum concentrations of ~90,000 ng/ml which may only be observed in some human subjects (such as fluorochemical production workers) exposed to extraordinarily high levels of environmental PFOS. This raises important issues regarding the broader pathophysiological/ clinical significance of the observed AIM2 inflammasome pathway in understanding the mechanisms by which chronic inflammatory disease and tissue dysfunction is induced by the much lower PFOS environmental exposure to consumer products in general human population cohorts.

Response: We have also established chronic exposure mouse model with low concentration of PFOS (i.p. 0.066 mg/kg per day for 30 days, which results in serum PFOS level as 86 ng/ml) that represent common exposure level of PFOS in human subjects. We found that wild type (WT) mice exposure to PFOS (0.066 mg/kg per day for 30 days) could also induce chronic inflammatory responses and tissue damage (see **New Figure 2a, related to Extended Data Figure 2a in the Supplemental material**). The proinflammatory cytokines IL-1 β , TNF- α and IL-6 were significantly increased in the serum, peritoneal fluids, liver, lung and kidney in the PFOS-treated mice, compared to those in control mice (see **New Figure 2b-d, related to Extended Data Figure 2b-d in the Supplemental material**). These results demonstrate that lower concentration of PFOS could also induce inflammatory cytokines production and tissue injury *in vivo*, which are in line with the acute exposure or asthma

exacerbation model (see **Extended Data Figure 1 in the Supplemental material and Figure 7 in the manuscript**).

We also treated THP-1-derived macrophages with suggested dose of PFOS (200 nM) in a time-dependent manner, and found that this concentration of PFOS could also induce pro-inflammatory cytokines release (IL-1 β , TNF- α , and IL-6) and cell death (see **New Figure 3a, related to Extended Data Figure 6a in the Supplemental material**). In addition, the much lower concentration range (10-200 nM) of PFOS could also trigger inflammatory cytokines production (IL-1 β , TNF- α , and IL-6) and cell death after PFOS exposure 48 h post stimulation (see **New Figure 3b, related to Extended Data Figure 6b in the Supplemental material**). Moreover, *AIM2* deficient THP-1 macrophages could significantly decrease IL-1 β production and cell death induced by PFOS (200 nM, 48 h) (see **New Figure 3c, related to Extended Data Figure 8e-f in the Supplemental material**). Taken together, the above *in vitro* cell system with low concentration of PFOS which representing the exposure level in general human populations, can also induce chronic inflammatory responses and tissue damage in animal and cell cultures via AIM2-dependent manner.

Specific Concerns (Experimental):

Comment 4: *The cell death analyses in control and AIM2-ko macrophages are limited to 6 hr PFOS treatments. Longer term treatments with PFOS should be included (particularly in the AIM2-ko cells) to analyze: 1) the kinetics of progression to secondary necrosis/lytic cell death; 2) roles of secondary GSDMD-independent but*

GSDME-dependent pyroptosis. (GSDME can be cleaved and activated by the apoptotic executioner caspase-3.)

Response: As shown in **Extended Data Figure 8 in the Supplemental material**, PFOS still induced obvious cell death in AIM2 inflammasome components deficient cells, suggesting that PFOS could trigger both pyroptosis and other types of cell death, including apoptosis. We further assayed the PFOS-induced apoptosis with Annexin V/7AAD staining by flow cytometry analysis, and observed that PFOS could trigger evident apoptosis in a dose-dependent manner (see **Extended Data Figure 11e in the Supplemental material**). These data indicated that PFOS treatment induces apoptosis. Previous studies reported that GSDME could be cleaved and activated by caspase-3³⁶⁻³⁸. We next sought to determine the participation of GSDME in the PFOS-induced AIM2-independent cell death by knocking down *GSDME* in the *AIM2* KO THP-1-derived macrophages (see **New Figure 10a, related to Extended Data Figure 11f in the Supplemental material**). However, we did not observe remarkable change of cell death between *GSDME* KD cells and control cells upon PFOS treatment (see **New Figure 10b, related to Extended Data Figure 11g in the Supplemental material**), suggesting that caspase-3-mediated cleavage of GSDME did not play a major role in the PFOS-induced AIM2-independent cell death. Hence, we speculated that other caspase-3 substrates than GSDME may be involved in AIM2-independent cell death. To confirm this hypothesis, we knocked down *caspase3* in *AIM2* KO THP-1-derived macrophages (see **New Figure 10a, related to Extended Data Figure 11f in the Supplemental material**), and found that the

decrease of AIM2-independent cell death by *caspase-3* knockdown appeared more pronounced than that by *GSDME* knockdown (see **New Figure 10b, related to Extended Data Figure 11g in the Supplemental material**). Together these results indicated that PFOS could induce caspase-3-mediated cell death and AIM2-dependent pyroptosis.

New Figure 10: a-b *GSDME* knockdown or *caspase-3* knockdown in *AIM2* KO THP-1-derived macrophages were treated with PFOS (150 μ M) as indicated time points. Cell lysates were collected to determine the knockdown efficiency of *GSDME* or *caspase-3* by immunoblot (**a**). Culture supernatant was then harvested to measure cell death by LDH release (**b**). In **a and b**, data are mean values \pm SEM, ns (non-significant), $P > 0.05$; ** $P < 0.01$; *** $P < 0.001$ (Student's t test).

Comment 5: Related to above concern, the cell death and AIM2 experiments are limited to macrophages. However, AIM2 inflammasome signaling and cell death in non-myeloid cells, particularly epithelial cells (PMID27846608; PMID29973713;

PMID31919014), can be a major driver of acute or chronic inflammatory responses to sterile stimuli. Experiments that analyze PFOS-induced AIM2 inflammasome signaling in an epithelial cell model should be included to establish whether PFOS drives similar AIM2 responses in non-myeloid cells.

Response: As the reviewer suggested, we treated human nasal epithelial progenitor cells (hNEPCs) with PFOS. The results showed that PFOS could significantly induced both the secretion of the proinflammatory cytokine IL-1 β and cell death in hNEPCs (see **New Figure 11a and b**). We then knocked down *AIM2* and found that PFOS-induced IL-1 β production and cell death were apparently decreased in *AIM2* KD hNEPCs (see **New Figure 11 c**), further confirming that PFOS could also induce *AIM2* inflammasome activation in non-myeloid cells. Consistently, in the bone-marrow transfer experiment, we found that despite its major role in the bone marrow-derived cells, *AIM2* might also function in other cell types such as epithelial cells, which contributes to the PFOS-induced tissue inflammation and damage.

New Figure 11: a-b hNEPCs were treated with PFOS as in a dose-dependent manner for 6 h or in time-dependent manner with 150 μ M PFOS. Cell supernatants were collected to measure IL-1 β release (**a**) and cell death (**b**). **c** AIM2 KD hNEPCs were treated with PFOS (150 μ M, 6 h). IL-1 β and LDH release was measured in the supernatants. Data are mean values \pm SEM, ns (non-significant), $P > 0.05$; ** $P < 0.01$; *** $P < 0.001$ (Student's t test).

Comment 6: Related to the above issue of non-myeloid AIM2 inflammasome signaling: The *in vivo* experiments described in Figures 6 and 7 shows that: 1) PFOS-induced tissue inflammation is suppressed in AIM2-deficient mice; and 2) PFOS-induced accumulation of IL-1 β , but not TNF- α or IL-6, is also suppressed in the AIM2-deficient mice. This raises very relevant questions as to whether production

of $IL-1\beta$ is, or is not, the major driver of inflammatory tissue damage in the PFOS-treated wildtype mice. Experiments using $IL-1\beta$ -knockout or $IL-1$ receptor-knockout mice are required to address this important mechanistic issue. It is possible that AIM2-dependent pyroptosis and/or AIM2-dependent $IL-18$ production, independently of AIM2-dependent $IL-1\beta$ accumulation, may be the major driver of inflammation in the PFOS-exposed mice.

Response: To further confirm that AIM2-dependent $IL-1\beta$ production is the major driver of PFOS-induced inflammation and tissue damage, WT mice or $Il-1\beta^{-/-}$ mice were i.p. injected with PFOS (25 mg/kg body weight per day for 5 days). We found that $Il-1\beta^{-/-}$ mice had reduced liver, lung and kidney inflammation and damage compared to WT mice (see **New Figure 12, related to Extended Data Figure 13a in the Supplemental material**), suggesting that AIM2-dependent $IL-1\beta$ production is the major driver of inflammation and tissue damage in the PFOS-exposed mice.

New Figure 12: Wild type (WT), $Il-1\beta^{-/-}$ mice (6 weeks old) were i.p. with PBS containing 2 % Tween-80 (Mock) or PFOS (25 mg/kg body weight per day for 5 days, dissolved in PBS containing 2 % Tween-80). At 5 days post-treatment, liver tissue,

lung tissue and kidney tissue of these mice were stained with hematoxylin-eosin (H&E) and assayed using a light microscope. Scale bar, 100 μm . The tissue (liver, lung, and kidney) injury score was determined in 5 randomly selected nonoverlapping fields from respective individual mouse tissue sections at x400 magnification. The scores were averaged in respective organs (liver, lung, and kidney) and individual animals. All histology analyses were conducted in a blinded manner. Data are mean values \pm SED, *** $P < 0.001$ (Student's t test).

Comment 7: *Figure 6b,c,d and Extended Data Figure 1b,c,d and Methods, lines 581-583: The 3 right-most subpanels in each of these figure panels and the relevant Methods section indicates: "At 5-days post-treatment mice liver, lung and kidney tissue were isolated and cultured for 24 hr, and the supernatants were analyzed by ELISA." More detail is required to explain how the liver, lung and kidney was processed for these ex vivo incubations and analyses. For example, were tissue slices prepared? If so, how thin were the slices? If not sliced, were crude cell suspensions prepared? If the latter, how were these cell suspensions generated? With tissue digesting enzymes?*

Response: As the reviewer suggested, we provided the detailed methods of tissue incubations as follows: the mice were sacrificed and then fully soaked them into 75% ethanol for 15 s to reduce the possibility of infection. Dissect and remove the entire liver, lung, and kidney into a petri dish containing 1% fetal bovine serum (FBS, Gibco) of sterile PBS, respectively. Cut out the gallbladder from the liver and wash the tissue using 1% FBS of sterile PBS for three times to ensure the tissue clear of blood. Cut the liver, lung, kidney into 1mm^3 , and digested by the medium A (sterile PBS, 1 M HEPES, 5 % KCL, 1 M glucose, 500 mM CaCl_2 , phenol red solution (Sigma). pH adjusted to 7.4 with 2 M NaOH) 0.2% IV-Collagenase (Sigma) in 37 $^\circ\text{C}$ for 15 min, respectively. Stop the digestion with DMEM medium (Hyclone) with 10 % (vol/vol) FBS. Filter the crude cell suspensions through gauze mesh filter (100 μm in diameter) and transfer the resulting cell suspensions into sterile tubes and centrifuge at 750 rpm

for 5 min. Discard the supernatant and repeat the wash procedure for three times via using medium A to resuspend the cell pellet and centrifuging at 750 rpm for 3 min. Resuspend the cells in cultured medium (DMEM medium containing 10% FBS) and check cells numbers and viability using hemocytometer and trypan blue. The cells are cultured into 60 mm tissue culture dishes via adding 5 ml of cell suspension with 1×10^7 cells per ml and softly rock the dish to ensure uniform distribution of cells. After culturing them for 24 h, cell supernatants are collected to determine the production of proinflammatory cytokines (IL-1 β , TNF- α and IL-6) by ELISA.

References

- 1 Chang ET, Adami HO, Boffetta P, Wedner HJ, Mandel JS. A critical review of perfluorooctanoate and perfluorooctanesulfonate exposure and immunological health conditions in humans. *Crit Rev Toxicol* **46**, 279-331 (2016).
- 2 Negri E, Metruccio F, Guercio V, Tosti L, Benfenati E, Bonzi R, et al. Exposure to PFOA and PFOS and fetal growth: a critical merging of toxicological and epidemiological data. *Crit Rev Toxicol* **47**, 482-508 (2017).
- 3 Dong GH, Tung KY, Tsai CH, Liu MM, Wang D, Liu W, et al. Serum polyfluoroalkyl concentrations, asthma outcomes, and immunological markers in a case-control study of Taiwanese children. *Environ Health Perspect* **121**, 507-13 (2013).
- 4 Bassler J, Ducatman A, Elliott M, Wen S, Wahlang B, Barnett J, et al. Environmental perfluoroalkyl acid exposures are associated with liver disease characterized by apoptosis and altered serum adipocytokines. *Environ Pollut* **247**, 1055-63 (2019).
- 5 Bevin E Blake, Susan M Pinney, Erin P Hines, Suzanne E Fenton, Kelly K Ferguson. Associations Between Longitudinal Serum Perfluoroalkyl Substance (PFAS) Levels and Measures of Thyroid Hormone, Kidney Function, and Body

- Mass Index in the Fernald Community Cohort. *Environ Pollut* **242**, 894-904 (2018).
- 6 Jin R, McConnell R, Catherine C, Xu S, Walker DI, Stratakis N, Jones DP, Miller GW, Peng C, Conti DV, Vos MB, Chatzi L. Perfluoroalkyl substances and severity of nonalcoholic fatty liver in Children: An untargeted metabolomics approach. *Environ Int* **134**:105220 (2020).
 - 7 Gaylord A, Trasande L, Kannan K, Thomas KM, Lee S, Liu M, Levine J. Persistent organic pollutant exposure and celiac disease: A pilot study. *Environ Res* **186**:109439 (2020).
 - 8 Yeung LW, Guruge KS, Taniyasu S, Yamashita N, Angus PW, Herath CB. Profiles of perfluoroalkyl substances in the liver and serum of patients with liver cancer and cirrhosis in Australia. *Ecotoxicol Environ Saf* **96**:139-46 (2013).
 - 9 Olsen GW, Zobel LR. Assessment of lipid, hepatic, and thyroid parameters with serum perfluorooctanoate (PFOA) concentrations in fluorochemical production workers. *Int Arch Occup Environ Health* **81**(2):231-46 (2007).
 - 10 Zhou Z, Shi Y, Vestergren R, et al. Highly elevated serum concentrations of perfluoroalkyl substances in fishery employees from Tangxun lake, china [J]. *Environmental science & technology*, **48** (7): 3864-74 (2014).
 - 11 Mo Yang, Li-Yue Li, Xiao-Di Qin, Xiao-Yan Ye, et al. Perfluorooctanesulfonate and perfluorooctanoate exacerbate airway inflammation in asthmatic mice and in vitro. *Science of the Total Environment*. In Press.
<https://doi.org/10.1016/j.scitotenv.2020.142365>
 - 12 Berthiaume J, Wallace KB. Perfluorooctanoate, perfluorooctanesulfonate, and N-ethyl perfluorooctanesulfonamido ethanol; peroxisome proliferation and mitochondrial biogenesis. *Toxicol Lett* **129**(1-2):23-32 (2002).
 - 13 Zhou X, Dong T, Fan Z, et al. Perfluorodecanoic acid stimulates NLRP3 inflammasome assembly in gastric cells. *Sci Rep* **7**:45468 (2017).
 - 14 Han R, Zhang F, Wan C, Liu L, Zhong Q, Ding W. Effect of perfluorooctane sulphonate-induced Kupffer cell activation on hepatocyte proliferation through

- the NF- κ B/TNF- α /IL-6-dependent pathway. *Chemosphere* **200**:283–294 (2018).
- 15 Olsen G W, Lange C C, Ellefson M E, et al. Temporal trends of perfluoroalkyl concentrations in American Red Cross adult blood donors, 2000-2010 [J]. *Environmental science & technology* **46** (11): 6330-8 (2012).
 - 16 Wu M, Sun R, Wang M, et al. Analysis of perfluorinated compounds in human serum from the general population in Shanghai by liquid chromatography-tandem mass spectrometry (LC-MS/MS) [J]. *Chemosphere* **168** (100-5) (2017).
 - 17 Dang EV, McDonald JG, Russell DW, Cyster JG. Oxysterol Restraint of Cholesterol Synthesis Prevents AIM2 Inflammasome Activation. *Cell* **171**(5):1057-1071.e11 (2017).
 - 18 Gan P, Gao Z, Zhao X, Qi G. Surfactin inducing mitochondria-dependent ROS to activate MAPKs, NF- κ B and inflammasomes in macrophages for adjuvant activity. *Sci Rep* 6:39303 (2016).
 - 19 Favreau DJ, Meessen-Pinard M, Desforges M, Talbot PJ. Human coronavirus-induced neuronal programmed cell death is cyclophilin d dependent and potentially caspase dispensable. *J Virol* **86**(1):81-93 (2012).
 - 20 Kumarswamy R, Seth RK, Dwarakanath BS, Chandna S. Mitochondrial regulation of insect cell apoptosis: evidence for permeability transition pore-independent cytochrome-c release in the Lepidopteran Sf9 cells. *Int J Biochem Cell Biol* **41**(6):1430-40 (2009).
 - 21 Oka N, Wang L, Mi W, Zhu W, Honjo O, Caldarone CA. Cyclosporine A prevents apoptosis-related mitochondrial dysfunction after neonatal cardioplegic arrest. *J Thorac Cardiovasc Surg* **135**(1):123-30, 130.e1-2 (2008).
 - 22 Lee YJ, Lee C. Porcine deltacoronavirus induces caspase-dependent apoptosis through activation of the cytochrome c-mediated intrinsic mitochondrial pathway. *Virus Res* **253**:112-123 (2018).
 - 23 Galluzzi L, Vanpouille-Box C. BAX and BAK at the Gates of Innate Immunity. *Trends Cell Biol* **28**(5):343-345 (2018).

- 24 McArthur K, Whitehead LW, Heddleston JM, Li L, Padman BS, Oorschot V, Geoghegan ND, Chappaz S, Davidson S, San Chin H, Lane RM, Dramicanin M, Saunders TL, Sugiana C, Lessene R, Osellame LD, Chew TL, Dewson G, Lazarou M, Ramm G, Lessene G, Ryan MT, Rogers KL, van Delft MF, Kile BT. BAK/BAX macropores facilitate mitochondrial herniation and mtDNA efflux during apoptosis. *Science* **359**(6378): eaao6047 (2018).
- 25 Riley JS, Quarato G, Cloix C, Lopez J, O'Prey J, Pearson M, Chapman J, Sesaki H, Carlin LM, Passos JF, Wheeler AP, Oberst A, Ryan KM, Tait SW. Mitochondrial inner membrane permeabilisation enables mtDNA release during apoptosis. *EMBO J* **37**(17): e99238 (2018).
- 26 Li S, Li H, Zhang YL, Xin QL, Guan ZQ, Chen X, Zhang XA, Li XK, Xiao GF, Lozach PY, Cui J, Liu W, Zhang LK, Peng K. SFTSV Infection Induces BAK/BAX-Dependent Mitochondrial DNA Release to Trigger NLRP3 Inflammasome Activation. *Cell Rep* **30**(13):4370-4385.e7 (2020).
- 27 Hu L, Chen M, Chen X, Zhao C, Fang Z, Wang H, Dai H. Chemotherapy-induced pyroptosis is mediated by BAK/BAX-caspase-3-GSDME pathway and inhibited by 2-bromopalmitate. *Cell Death Dis* **11**(4):281(2020).
- 28 Zhou R, Yazdi AS, Menu P, Tschopp J. A role for mitochondria in NLRP3 inflammasome activation. *Nature* **469**(7329):221-5 (2011).
- 29 Zhong Z, Liang S, Sanchez-Lopez E, He F, Shalapour S, Lin XJ, et al. New mitochondrial DNA synthesis enables NLRP3 inflammasome activation. *Nature* **560**,198-203 (2018).
- 30 Hu B, Jin C, Li HB, Tong J, Ouyang X, Cetinbas NM, et al. The DNA-sensing AIM2 inflammasome controls radiation-induced cell death and tissue injury. *Science* **354** (2016).
- 31 Sharma BR, Karki R, Kanneganti TD. Role of AIM2 inflammasome in inflammatory diseases, cancer and infection. *Eur J Immunol* **49**, 1998-2011 (2019).
- 32 Bae JH, Jo SI, Kim SJ, Lee JM, Jeong JH, Kang JS, Cho NJ, Kim SS, Lee EY, Moon JS. Circulating Cell-Free mtDNA Contributes to AIM2

- Inflammasome-Mediated Chronic Inflammation in Patients with Type 2 Diabetes. *Cells* **8**(4):328 (2019).
- 33 Dostert C, Pétrilli V, Van Bruggen R, Steele C, Mossman BT, J, T. (2008). Innate immune activation through Nalp3 inflammasome sensing of asbestos and silica. *Science* **320**, 674-7.
- 34 Hornung V, Bauernfeind F, Halle A, Samstad EO, Kono H, Rock KL, et al. Silica crystals and aluminum salts activate the NALP3 inflammasome through phagosomal destabilization. *Nat Immunol* **9**, 847-56 (2008).
- 35 Muñoz-Planillo R, Kuffa P, Martínez-Colón G, Smith BL, Rajendiran TM, Núñez G.. K⁺ efflux is the common trigger of NLRP3 inflammasome activation by bacterial toxins and particulate matter. *Immunity* **38**, 1142-53 (2013).
- 36 Wang Y, Gao W, Shi X, Ding J, Liu W, He H, Wang K, Shao F. Chemotherapy drugs induce pyroptosis through caspase-3 cleavage of a gasdermin. *Nature* **547**(7661):99-103 (2017).
- 37 Yu J, Li S, Qi J, Chen Z, Wu Y, Guo J, Wang K, Sun X, Zheng J. Cleavage of GSDME by caspase-3 determines lobaplatin-induced pyroptosis in colon cancer cells. *Cell Death Dis* **10**(3):193 (2019).
- 38 Zhang Z, Zhang Y, Xia S, Kong Q, Li S, Liu X, Junqueira C, Meza-Sosa KF, Mok TMY, Ansara J, Sengupta S, Yao Y, Wu H, Lieberman J. Gasdermin E suppresses tumor growth by activating anti-tumor immunity. *Nature* **579**(7799):415-420 (2020).

REVIEWERS' COMMENTS:

Reviewer #1 (Remarks to the Author):

It would be helpful to include some information about the structural differences between two different PFAS with variable activity towards IL-1 production.

Reviewer #2 (Remarks to the Author):

I appreciate the authors extensive efforts to address my major concern, i.e. the surprising link between PFAS induced MPTP (largely supported through use of CSA in the initial ms) and BAX/BAK activity. As I stated this was unexpected given that BAX/BAK dependent MOMP does not invoke MPTP, equally MPTP (leading to mitochondrial rupture) does not require BAX or BAK.

The data presented here, the authors propose that BAX interaction with cypD is responsible for this cross-talk. While they nicely show an interaction between BAX and cypD, given that the location of these proteins is well-established as being matrix (cypD) and cytosolic or mitochondrial outer membrane (BAX), its very difficult to reconcile how these proteins would interact in a physiological manner since they are spatially separated from one another (by virtue of the mitochondrial inner membrane) - a major concern is that the interaction is a post-lysis artefact of the ip method.

Reconciling the data (both old/new) - I would propose that an equally feasible model is that rather than inducing MPTP (which is not actually demonstrated anywhere in this study), PFAS activity leads to BAX/BAK mediated MOMP (associated mtDNA release, cell death) in a CYPD dependent (but MPTP independent manner). I would suggest the authors, address alternative models (textually) in any revised ms.

Reviewer #3 (Remarks to the Author):

General Comments: The authors have performed many new experiments that effectively addressed the concerns and recommendations raised by the three referees. The revised MS now provides a significant and mechanistically informative contribution to the literature linking various environmental pollutants (chemical or particulate) to the induction of sterile inflammasome signaling and its multiple proinflammatory consequences in vivo. Although no new experiments are required, the text of the revised MS still needs some editing, clarification and extension.

Specific Comments:

1. lines 87-88, 484: Although environmental particulates such as silica and asbestos do induce NLRP3 inflammasome that correlates with cytosolic accumulation of cathepsin B (cath B), recent studies have indicated that cathB per se is not a direct activator of NLRP3. Rather, lysosomal disruption by the internalized particulates induces release of multiple lysosomal cathepsins (and other proteases) which trigger perturbation of plasma membrane permeability (via unclear mechanisms), increased K⁺ efflux and consequent NLRP3 activation. The authors should merely indicate that lysosomal disruption by environmental particulates drives NLRP3 inflammasome activation without specifying roles for cathepsin B.

2. lines 274, 280-281, 319-320: These parts of the text still contain some imprecise descriptions of mitochondrial cytochrome C location and the mechanisms of its release into the cytosol during intrinsic apoptosis. As the authors correctly noted in their replies to the reviewer comments, cyto C is an "inter" membrane protein, not an "inner" membrane protein. However, line 274 of the revised MS still describes cyto C as an "inner" membrane proteins. As a component of the mitochondrial inter membrane compartment, cyto C is predominantly by the BAX/BAK pores that mediate MOMP, rather than MPTP. This needs to be clarified in lines 280-281 and 319-320.

3. Lines 416, 425, 434, 445 of the revised discussion: Some type of error message is noted in lines that cite new references.

4. Methods: The MS now provides important new data showing that increased cytosolic Ca²⁺ mediates both a PKC > NFκB axis (as part of the AIM2 inflammasome priming phase), and a PKC> JNK> BAX/BAK axis (as part of the MOMP-mediated mtDNA release phase). However, there is no description of the methods for the Fluo4-based measurements of Ca²⁺ fluxes in the PFOS-stimulated cells.

5. Regarding these changes in PFOS-induced Ca²⁺ release from the ER, the authors should provide some discussion of the potential signaling mechanisms that underlie this Ca²⁺ release response,.

Appendix-I

Reviewer #1 (Remarks to the Author):

It would be helpful to include some information about the structural differences between two different PFAS with variable activity towards IL-1 production.

Response: The structural difference between PFOA and PFOS is shown in **New Fig.1.**

In previous response letter, we demonstrated that PFOS but not PFOA could induce IL-1 β production, and Ca²⁺-dependent pathway was critical for PFOS-induced IL-1 β production. Thus, in our study, we focus on PFOS but not PFOA (only showed in the last response letter but not the main manuscript).

New Fig 1. The chemical structural formula of PFOS and PFOA.

Reviewer #2 (Remarks to the Author):

I appreciate the authors extensive efforts to address my major concern, i.e. the surprising link between PFAS induced MPTP (largely supported through use of CSA in the initial ms) and BAX/BAK activity. As I stated this was unexpected given that BAX/BAK dependent MOMP does not invoke MPTP, equally MPTP (leading to mitochondrial rupture) does not require BAX or BAK.

The data presented here, the authors propose that BAX interaction with cypD is responsible for this cross-talk. While they nicely show an interaction between BAX and cypD, given that the location of these proteins is well-established as being matrix

(cypD) and cytosolic or mitochondrial outer membrane (BAX), its very difficult to reconcile how these proteins would interact in a physiological manner since they are spatially separated from one another (by virtue of the mitochondrial inner membrane) - a major concern is that the interaction is a post-lysis artefact of the ip method.

Reconciling the data (both old/new) - I would propose that an equally feasible model is that rather than inducing MPTP (which is not actually demonstrated anywhere in this study), PFAS activity leads to BAX/BAK mediated MOMP (associated mtDNA release, cell death) in a CYPD dependent (but MPTP independent manner). I would suggest the authors, address alternative models (textually) in any revised ms.

Response: As the reviewer suggested, we have modified our descriptions of indicated findings and added and discussed the suggested model in the Discussion part on pages 22 to 23.

Reviewer #3 (Remarks to the Author):

General Comments: The authors have performed many new experiments that effectively addressed the concerns and recommendations raised by the three referees. The revised MS now provides a significant and mechanistically informative contribution to the literature linking various environmental pollutants (chemical or particulate) to the induction of sterile inflammasome signaling and its multiple proinflammatory consequences in vivo. Although no new experiments are required, the text of the revised MS still needs some editing, clarification and extension.

Specific Comments:

1. lines 87-88, 484: Although environmental particulates such as silica and asbestos do induce NLRP3 inflammasome that correlates with cytosolic accumulation of cathepsin B (cath B), recent studies have indicated that cathB per se is not a direct activator of NLRP3. Rather, lysosomal disruption by the internalized particulates induces release of multiple lysosomal cathepsins (and other proteases) which trigger perturbation of plasma membrane permeability (via unclear mechanisms), increased K⁺ efflux and consequent NLRP3 activation. The authors should merely indicate that lysosomal disruption by environmental particulates drives NLRP3 inflammasome activation without specifying roles for cathepsin B.

Response: We thank the reviewer for noticing this and changed the indicated description as “silica crystals could induce lysosomal disruption, which then drives NLRP3 inflammasome activation, and leads to lung inflammation” on page 4.

2. lines 274, 280-281, 319-320: These parts of the text still contain some imprecise descriptions of mitochondrial cytochrome C location and the mechanisms of its release into the cytosol during intrinsic apoptosis. As the authors correctly noted in their replies to the reviewer comments, cyto C is an "inter" membrane protein, not an "inner" membrane protein. However, line 274 of the revised MS still describes cyto C as an "inner" membrane protein. As a component of the mitochondrial intermembrane compartment, cyto C is predominantly by the BAX/BAK pores that mediate MOMP, rather than MPTP. This needs to be clarified in lines 280-281 and 319-320.

Response: We thank the reviewer for noticing this and changed the description accordingly in the manuscript.

3. Lines 416, 425, 434, 445 of the revised discussion: Some type of error message is noted in lines that cite new references.

Response: We have changed it accordingly.

4. Methods: The MS now provides important new data showing that increased cytosolic Ca²⁺ mediates both a PKC > NFkB axis (as part of the AIM2 inflammasome priming phase), and a PKC> JNK> BAX/BAK axis (as part of the MOMP-mediated mtDNA release phase). However, there is no description of the methods for the Fluo4-based measurements of Ca²⁺ fluxes in the PFOS-stimulated cells.

Response: We thank the reviewer for noticing this and add the protocol of “Cytosolic Ca²⁺ detection” in the Method part.

5. Regarding these changes in PFOS-induced Ca²⁺ release from the ER, the authors should provide some discussion of the potential signaling mechanisms that underlie this Ca²⁺ release response.

Response: As the reviewer suggested, we have added and discussed the mechanisms that underlie PFOS-induced Ca²⁺ release from the ER in the Discussion part on page 23.